# Multi-Agent Collaborative Data Selection for Efficient LLM Pretraining

## Abstract

Efficient data selection is crucial to accelerate the pretraining of large language models (LLMs). While various methods have been proposed to enhance data efficiency, limited research has addressed the *inherent conflicts* between these approaches to achieve optimal data selection for LLM pretraining. To tackle this problem, we propose a novel *multi-agent collaborative data selection* mechanism. Each data selection method independently prioritizes data based on its specific criterion and updates its prioritization rules using the current state of the model, functioning as an independent agent for data selection. Additionally, an agent console is designed to adjust the impacts of different agents at various stages and dynamically integrate information from all agents throughout the LLM pretraining process. We conduct extensive empirical studies to evaluate our multi-agent framework. The experimental results demonstrate that our approach significantly improves data efficiency, accelerates convergence in LLM pretraining, and achieves an average performance gain up to $10.5\%$ across multiple language model benchmarks compared to the state-of-the-art methods.

## 1 Introduction

Efficient data selection is crucial for the pretraining of large language models (LLMs), as the quality of training data significantly impacts the statistical efficiency of the training procedure and the model performance (Brown, 2020; Du et al., 2022; Chowdhery et al., 2023). Recently, we have witnessed numerous approaches, such as filtering high-quality data (Xie et al., 2023b; Wettig et al., 2024), mixing data from multiple domains (Xie et al., 2023a; Liu et al., 2024), and selecting data that optimally boosts downstream task performance dynamically (Engstrom; Yu et al., 2024), which aim to improve data efficiency by prioritizing more informative training samples. However, these methods often operate independently or in isolated settings, limiting their potential when integrated into a collaborative framework. In this work, we want to explore *how to effectively, flexibly, and robustly combine these advanced data selection techniques through the dynamic pretraining process*, addressing the challenges of optimizing data efficiency for LLM pretraining at scale.

Nowadays, various heuristic methods have been proposed to provide measurements for the data samples used during LLM pre-training, aiming to optimize data efficiency by selecting or weighting the most informative training examples. However, we observe that integrating multiple data selection and mixing strategies presents significant challenges due to their *inherent conflicts*. For example, high-quality data identified by scoring functions may not align with data that strongly impact model performance as measured by influence functions (Engstrom); similar conflicts also exists between other methods — further details are enumerated in §2. These observations actually motivate us to launch a systematic discussion about how to effectively integrate these methods during the dynamic pretraining process that provides superior data efficiency for LLM pretraining.

On the other hand, effectively integrating these data selection methods into a single framework is much harder to implement than to ask for. In fact, one may have to explore an exponential space to find the optimal combination for different data sampling schemas. Such a heavy burden will be further amplified when we consider the dynamic adjustment during the training process introduced by state-of-the-art online data selection approaches. In fact, different from the offline methods that leverage *fixed* classifier-based scoring functions (Brown, 2020; Gao et al., 2020; Du et al., 2022; Chowdhery et al., 2023; Sachdeva et al., 2024; Wettig et al., 2024), domain weights (Brown, 2020;

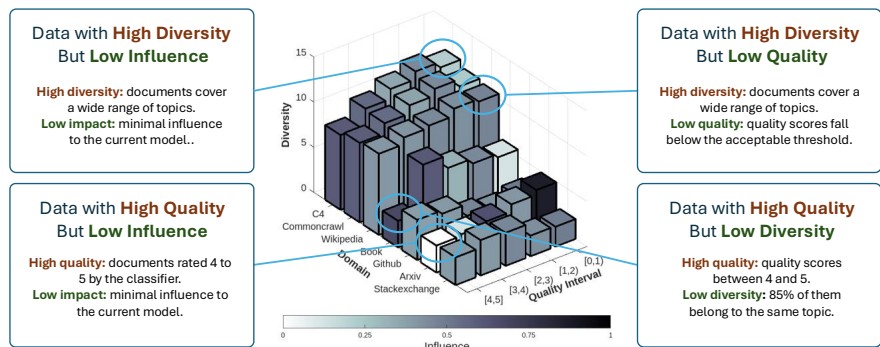

Figure 1: **Statistics of the SlimPajama dataset.** This figure shows the distribution of overall 627B tokens data across four dimensions: quality, domain, topic diversity, and impact on the pretrained model at the 1500th step. Each bar represents a subset defined by a specific quality interval and domain. This visualization reveals the conflict among diversity, quality, and model influence.

Team, 2024; Rae et al., 2021; Xie et al., 2023a; Liu et al., 2024), or down-sampled topics (Team, 2024; Chen et al., 2024), online methods use techniques like influence functions (Engstrom; Yu et al., 2024) to assess the model's sensitivity to individual data points during the LLM pretraining process to enable dynamic selection of high-impact data at any optimization step. Such an online paradigm is computationally intensive by itself, which presents the essential challenge for an effective dynamic integration in our problem setting that demands a computationally efficient solution to preserve or even amplify the advantage of each data selection heuristic.

To address these challenges, we conduct a case study to identify the inherent conflicts for existing data selection methods and provide a multi-agent collaborative data selection framework to resolve this issue. Our multi-agent framework is inspired by the classical definition of intelligence outlined by (Russell & Norvig, 2016), where an agent is defined as an entity that perceives a state and maps the observed state to actions. In our approach, data selection is achieved through the collaboration of these agents. Concretely, we make the following contributions:

**Contribution 1:** In §2, we present a case study on the SlimPajama dataset, revealing intriguing relationships among four widely used data selection metrics in LLM pretraining: data quality, topic diversity, data impact, and data domain. The analysis highlights inherent conflicts among these methods, yet studies (Wettig et al., 2024; Xie et al., 2023a; Yu et al., 2024) show that using any single metric as a standalone data selection criterion can still produce effective trajectories for convergence. This underscores the need to understand and effectively integrate these differing approaches.

**Contribution 2:** We propose a novel multi-agent collaborative data selection mechanism in §3. In this framework, each data selection method operates as an agent capable of providing scores to prioritize the training data samples. We also design an agent console to effectively integrate the scores from all agents, producing optimized data selection results. Furthermore, we implement a dynamic collaboration mechanism, where the contribution of each agent can be adjusted dynamically throughout the LLM training process. This approach enables more flexible and adaptive data selection, improving overall data efficiency during LLM pretraining.

**Contribution 3:** To evaluate our multi-agent collaborative data selection method, we conducted extensive experiments to show that: (1) In end-to-end experiments, our approach significantly improves data efficiency, leading to faster convergence in LLM training and achieving an average improvement up to $10.5\%$ across various language model benchmarks compared to baseline methods (§4.1); and (2) Ablation studies confirm that the design and implementation of key components in our multi-agent framework are crucial for attaining this advanced performance (§4.2). These findings highlight the effectiveness of our method to optimize data efficiency for LLM training.

## 2 CASE STUDY - INHERENT CONFLICTS IN DATA SELECTION

In this section, we present several observations derived from the SlimPajama datasets (Soboleva et al., 2023), which reveal some inherent conflicts for different data selection measurements. To

conduct this case study, we first label all data from the SlimPajama datasets using the quality scorer FineWeb-Edu (Lozhkov et al., 2024). We then divide the data into subsets based on domain and quality ranges. From each subset, we uniformly sample data to assess topic diversity, i.e., the topic classification of the sampled data according to our methods. We analyze this diversity by examining the topic distribution within each subset. Additionally, we compute the normalized influence of the data on a pretrained 1.3B model at the 1500th step using influence functions to evaluate the data's impact on the model (Engstrom). Figure 1 illustrates the results, which presents a bar chart representing four dimensions: quality, domain, topic diversity, and influence on the pretrained model. The **x-axis** shows data quality, with higher intervals reflecting better scores from the FineWeb-Edu quality scorer. The **y-axis** indicates the dataset's domain, while the **z-axis** shows topic diversity within each subset, with taller bars indicating more diversity. The **color gradient** represents influence on the model, with darker shades showing greater impact. From this analysis, we highlight the following interesting observations:

- *High-quality data identified by the quality scorer may not have a significant impact on model performance.* For example, ArXiv documents rated between 4 and 5 by the scorer are considered high-quality. However, at the 1500th training step, they exert minimal impact on the model according to the influence functions, revealing a discrepancy between data quality and model impact. This observation is consistent with the previous discussion in Engstrom.
- *High-quality data may exhibit low topic diversity.* Documents in the Book domain with a quality score of 4 to 5 are classified as high-quality by the scorer. Nevertheless, 85% of these documents belong to the same topic, indicating a lack of diversity.
- *Data with high topic diversity may not strongly influence model performance.* Documents from the C4 domain display considerable topic diversity. However, at the 1500th training step, they have limited impact on the model as measured by the influence functions, suggesting a conflict between diversity and model influence.
- *Data with high topic diversity can be low quality.* Wikipedia documents show substantial topic diversity, which benefits the topic classifier. However, some of these documents are rated as low-quality by the quality classifier, revealing a trade-off between diversity and quality.

We believe this inherent conflict illustrates that a naive ensemble of these mechanisms may lead to poor performance in terms of data efficiency for LLM pretraining, which motivates the design and implementation of our multi-agent collaborative framework in §3.

## 3 MULTI-AGENT COLLABORATIVE DATA SELECTION

In this section, we present the formalization of the data selection problem in §3.1, outline the overall framework of our methods in §3.2, and detail the agent initialization and update in §3.3, along with the collaborative mechanism in §3.4.

### 3.1 PROBLEM FORMULATION

We follow the definition of the data selection problem in Engstrom and Yu et al. (2024) with slight modification. The objective for data selection is to choose a subset of size $k$ from the entire pretraining dataset in such a way that the trained model's loss on downstream tasks is minimized. Let $\mathcal{O}$ represent an optimization algorithm that maps a training dataset to a trained model. The optimal subset $\mathcal{D}_k^*$ of the pretraining dataset $\mathcal{D}$ can be expressed as:

$$\mathcal{D}_k^* \coloneqq \underset{\mathcal{D}_k \subset \mathcal{D}, |\mathcal{D}_k| = k}{\arg\min} \mathcal{L}(\mathcal{D}_k \mid \mathcal{M}, \mathcal{T}_{\text{eval}}), \tag{1}$$

where $\mathcal{L}(\mathcal{D}_k \mid \mathcal{M}, \mathcal{T}_{\text{eval}}) \coloneqq \mathbb{E}_{x \sim \mathcal{T}_{\text{eval}}} [\ell(x; \mathcal{O}(\mathcal{M}, \mathcal{D}_k))]$ denotes the loss (e.g., cross-entropy loss) for model $\mathcal{M}$ on example $x$ of the downstream task $\mathcal{T}_{\text{eval}}$. Minimizing this objective directly is computationally challenging. Given that the real downstream tasks are unknown during model training, prior works have approximately optimized this problem by minimizing the loss on selected reference tasks $D_{\text{ref}}$ (e.g. LAMBADA Paperno et al. (2016), SQuAD Rajpurkar (2016) and Jeopardy Tunguz (2019) in Engstrom). Specifically, they train proxy models to compute one-step training loss (Yu et al., 2024) or influence functions (Engstrom) on the reference tasks to approximate the true loss. However, this approach heavily depends on the selection of the reference tasks, while the chosen reference tasks may not be fully representative of all potential downstream tasks.

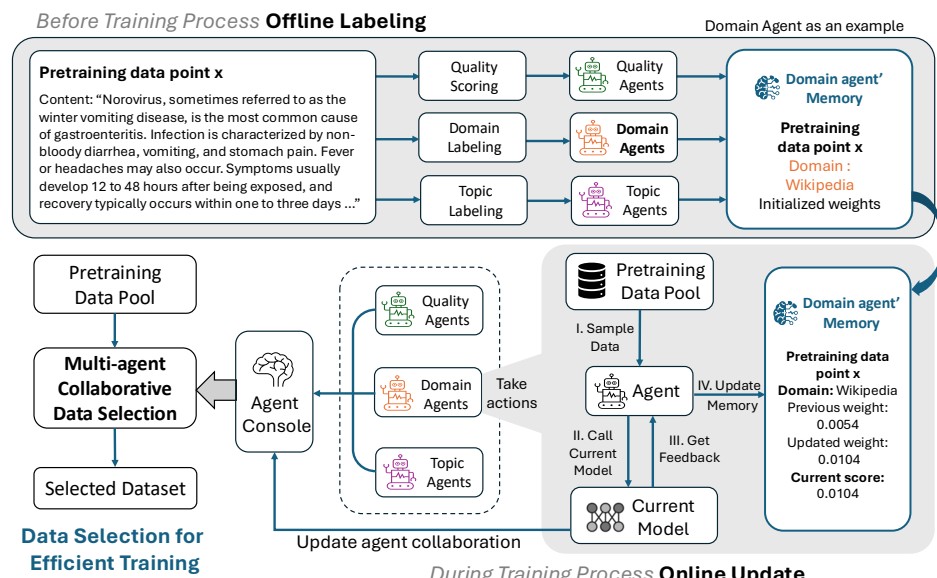

Figure 2: **Illustration of multi-agent collaborative framework.** Multi-agent collaborative framework for pretraining data selection that integrates multiple perspectives by combining offline priors and online model-derived preferences.

To avoid this obstacle, we do not directly minimize the loss on the reference tasks. Instead, we view this loss as a *reward signal* that guides the update of predefined data selection methods. Concretely, we define a reward function $R(\mathcal{D}_k \mid \mathcal{M}, \mathcal{T}_{\text{ref}})$, where the reward is based on the performance gain of current model $\mathcal{M}$ trained on the subset $\mathcal{D}_k$ and evaluated on the reference tasks $\mathcal{T}_{\text{ref}}$. Then our optimization goal becomes maximizing this reward over time, as:

$$\mathcal{D}_k^* = \underset{\mathcal{D}_k \subset \mathcal{D}, |\mathcal{D}_k| = k}{\arg\max} \; \mathbb{E}\left[R(\mathcal{D}_k \mid \mathcal{M}, \mathcal{T}_{\text{ref}})\right],$$
$$\text{where } R(\mathcal{D}_k \mid \mathcal{M}, \mathcal{T}_{\text{ref}}) := \mathbb{E}_{x \sim \mathcal{T}_{\text{ref}}}\left[-\ell(x; \mathcal{O}(\mathcal{M}, \mathcal{D}_k))\right]. \tag{2}$$

### 3.2 MULTI-AGENT COLLABORATIVE DATA SELECTION FRAMEWORK

In order to solve the optimization problem in Equation 2, we develop a framework illustrated in Figure 2. This framework consists of two primary stages: the offline labeling stage and the online update stage. Before the training process, some initial information (i.e., the initialized measurements in some heuristic) is computed for the entire pretraining corpus, and this information is stored separately in each agent's memory (formally defined below). During the training process, the current model (i.e., LLM to be trained) is used to update the agents' memory and their collaboration mechanism based on rewards computed on the current model. An agent console is responsible for aggregating the opinions of each agent and making the final data selection decision. Formally, we define the agent in Definition 1 and the agent console in Definition 2. Detailed formulation is in Appendix A.3.

**Definition 1** (Agent). An *agent* $\mathcal{A}$ is a data selection method defined by a specific attribute (e.g., quality, domain, or topic) with memory $\mathcal{H}_\mathcal{A}$ that stores labels for each data point and their associated scores. During training, the agent takes several actions: (1) Sample data $\mathcal{D}_\mathcal{A}$ according to predefined sampling distribution, (2) Call the current model to compute the reward $\mathcal{R}(x_i)$ for each sample $x_i \in \mathcal{D}_\mathcal{A}$, (3) Get feedbacks from current model state, and (4) Update the internal weights $\mathbf{w}_\mathcal{A}$ in its memory. Then it assigns a score $\mathcal{S}_\mathcal{A}$ to each data point based on its updated memory, prioritizing the good data according to the updated weights. One agent's objective is to maximize its reward by updating this agent's internal weights and increasing the score of higher-reward data points.

**Definition 2** (Agent Console). The *agent console* is in charge of coordinate opinions from different agent to make final decision of selecting dataset for next training stage. Specifically, it consolidates

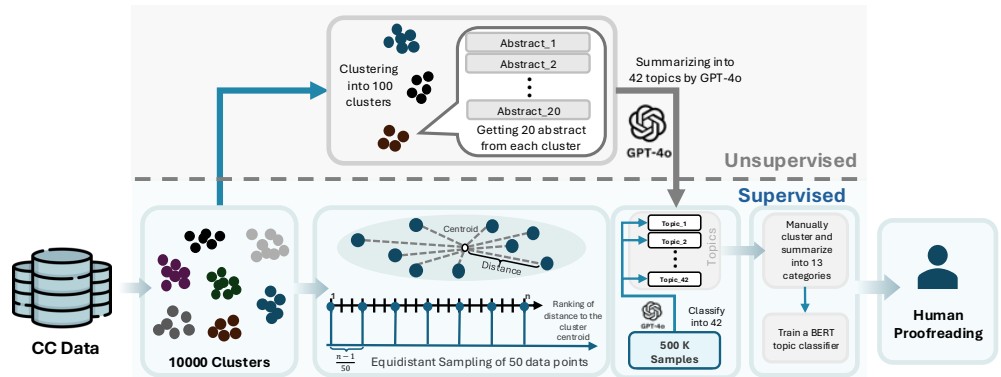

Figure 3: **Illustration of training process for topic classifier.** This diagram shows the process of training a BERT-based topic classifier using CommonCrawl data. 1.44 billion documents are clustered to generate topics. GPT-4o handles topic summarization and annotation, while a BERT model is trained to classify 13 topics, with humans doing final proofreading.

scores $\mathcal{S}_{\mathcal{A}}(x_i)$ from multiple agents $\{\mathcal{A}_1, \ldots, \mathcal{A}_n\}$ to calculate a final score $S(x_i)$ for each data point $x_i$, and takes final action of selecting dataset. The console adjusts the collaborative weights $\theta_{\mathcal{A}}$ for each agent based on their respective aggregate rewards $\mathcal{R}_{\mathcal{A}}$, balancing their contributions during training. In cases where there are conflicts in the decisions made by agents, the console resolves these by adjusting the weights $\theta_{\mathcal{A}}$ to prioritize the agents that have a greater positive impact on the model's performance, ensuring an effective data selection process.

Now the reward signal is actually came from multiple agents, the optimization goal in Equation 2 becomes maximizing the expectation of collaborative agents $\mathbb{E}_{\{\mathcal{A}_1, \ldots, \mathcal{A}_n\}}\left[R(\mathcal{D}_k \mid \mathcal{M}, \mathcal{T}_{\text{ref}})\right]$. In our current implementation, we include three agents, which are topic agents, quality agents and domain agents. They are aiming to maximize the rewards from topic, quality and domain perspective respectively. In the following sections, we will detail how we initialize and update a single agent (§3.3), and how we update the agent console for multi-agent collaboration (§3.4).

### 3.3 SINGLE AGENT INITIALIZATION AND UPDATE

**Agent initialization.** As defined in Definition 1, for a particular agent, we have to maintain its memory $\mathcal{H}_{\mathcal{A}}$ throughout the training process. Before training process begin, we label the whole training dataset $\mathcal{D}$ offline and store the labeled information to the memory of corresponding agents. Specifically, for each data point $x_i \in \mathcal{D}$, i.e., a single document in our settings, we first get the quality, topic and domain label using scorer and classifier.

For *quality agent*, we adopt the FineWeb-Edu quality scorer (Lozhkov et al., 2024), which is fine-tuned as a BERT-like regression model Merrick et al. (2024) using Llama3-70B-Instruct annotated 500k examples. This will give out a successive quality score $\text{Quality}(x_i) \in \mathcal{R}^{[0,5]}$ with higher score represent higher quality. We then map this score into five quality intervals $\{I_j\}_{j=1}^5$, as

$$\text{Quality}(x_i) \mapsto I_j = \begin{cases} [j-1, j), & \text{if } \text{Quality}(x_i) \in [j-1, j) \quad (j = 1, 2, 3, 4) \\ [4, 5], & \text{if } \text{Quality}(x_i) \in [4, 5] \quad (j = 5) \end{cases} \tag{3}$$

We store the quality interval corresponding to each data point in the quality agent's memory.

For the *domain agent*, we use the document's meta-information, label the data with domain information and save this into the domain agent's memory, where the domain $\text{Domain}(x_i)$ belongs to one of ArXiv, Book, Wikipedia, CommonCrawl, GitHub, StackExchange, C4.

For the *topic agent*, due to the absence of a suitable pretrained model for topic classification and labeling, we designed a classification schema using 1.44 billion documents collected by the Common Crawl project (Project, 2007) and fine-tuned a BERT-like regression model on 500k GPT-4o annotated samples, the overall pipeline is depicted in Figure 3. Further details on the topic classification

approach and BERT model training are provided in §A.1. Using this topic classifier, we categorize each document into one of 13 topics: Activity, Education, Entertainment, Finance, Health, Business and Industrial, Infrastructure, Literature and Art, Nature, Others, Law and Government, Networking, Technology, and store the topic information in the topic agent's memory.

We initialize the weight of the topic agent and domain agent following the RegMix Liu et al. (2024) framework. Unlike the original RegMix, which only considers mixing data based on domain labels, we examine data mixing weights based on domain as well as the topic labels. We initialize our quality agent similar to the data selection decision of QuRating Wettig et al. (2024) and FineWeb-Edu Lozhkov et al. (2024). Further details can be found in §A.7.3. The initial weights for each agent are stored in their respective memory.

During the training phase, we leverage the current model to adjust the weight of each agent. As depicted in Figure 2, at the data selection stage, each agent performs several actions to update its memory and inform decision-making. Take domain agent as an example, it takes three-step action during data selection stage: (1) Sample pretraining data points from the pretraining data pool, distributing them uniformly across each domain; (2) Call the current model to assess the reward of each data point and gather feedback; (3) Update the memory of domain weights based on gathered feedback and adjusts the score for each data point by incorporating prior knowledge from the offline labeling process. This process is similarly followed by the quality agent and the topic agent.

**Agent update.** After sampling data uniformly from agent search space, each agent updates its internal weights using local information based on the sampled data. For example, for domain agent, it calculates the average influence of each domain. For agent $\mathcal{A} \in \{\mathcal{A}_{\text{Quality}}, \mathcal{A}_{\text{Domain}}, \mathcal{A}_{\text{Topic}}\}$, it updates its internal weights by calculating the overall rewards sampled from each interval as:

$$\overline{R}_{\mathcal{A}}^{j} = \frac{1}{|D_{\mathcal{A}}^{j}|} \sum_{x_i \in D_{\mathcal{A}}^{j}} \mathcal{R}(x_i \mid \mathcal{M}, \mathcal{T}_{\text{ref}}), \tag{4}$$

where $D_{\mathcal{A}}^{j}$ represents the sample set of the $j$-th subcategory under agent $\mathcal{A}$, e.g. Wikipedia for domain agent. And $x_i$ is a sample within this sample set. Then, the sliding averaging is used to update the weight for each subcategory $w_{\mathcal{A}}^{j}$ with current rewards:

$$w_{\mathcal{A}}^{j} \leftarrow (1 - \eta_A) \cdot w_{\mathcal{A}}^{j} + \eta_A \cdot \overline{R}_{\mathcal{A}}^{j}, \tag{5}$$

where $\eta_{\mathcal{A}}$ is the sliding average factor to tradeoff bias-variance. The overall updated weight of agent $\mathcal{A}$ is an vector in $n_{\mathcal{A}}$ dimension, $\mathbf{w}_{\mathcal{A}} = [w_{\mathcal{A}}^{1}, ..., w_{\mathcal{A}}^{n_{\mathcal{A}}}]$, where $n_{\mathcal{A}}$ is the number of total subcategory within the space of agent $\mathcal{A}$. Utilizing the prior memory stored by each agent, it can give out a final score for each data point as $\mathcal{S}_{\mathcal{A}}(x_i) = w_{\mathcal{A}}^{j}$, where $j$ is the subcategory that $x_i$ belongs to.

**Calculating the rewards.** For each sampled data point, we approximate rewards using influence functions Engstrom; Yu et al. (2024). The influence function value is computed to measure the impact of each sample on the model's performance. The formula for the influence function is:

$$\begin{aligned} \mathcal{R}(x_i \mid \mathcal{M}, \mathcal{T}_{\text{ref}}) &= \hat{\mathbb{E}}_{x \sim \mathcal{T}_{\text{ref}}} \left[ -\ell(x; \mathcal{O}(\mathcal{M}, \mathcal{D}_k)) \right], \\ &= -\nabla_{\mathcal{M}} \mathcal{L}(\mathcal{T}_{\text{ref}} \mid \mathcal{M})^{\top} H_{\mathcal{M}}^{-1} \nabla_{\mathcal{M}} \mathcal{L}(x_i \mid \mathcal{M}), \end{aligned} \tag{6}$$

where $H_{\mathcal{M}} = \frac{1}{n} \sum_{i=1}^{n} \nabla_{\mathcal{M}}^{2} \mathcal{L}_{\mathcal{M}}(x_i \mid \mathcal{M})$ is the Hessian and its positive definite. Details of calculating influence functions for pretraining data point can be found in §A.9.

## 3.4 MULTI-AGENT COLLABORATION

Ultimately, the agent console defined in Definition 2 aggregates all agents' feedback to compute a final score for each data point, determining the final data selection decision.

**Multi-agent collaboration.** In the context of multi-agent collaboration, the weighted score for each agent must be calculated to evaluate their respective contributions effectively. This calculation takes into account various factors specific to each agent. For every data sample $x_i$, the overall score $S(x_i)$ is determined by the following formula:

$$S(x_i) = (\theta_{\text{Quality}} \cdot \mathcal{S}_{\text{Quality}}(x_i) + \theta_{\text{Domain}} \cdot \mathcal{S}_{\text{Domain}}(x_i) + \theta_{\text{Topic}} \cdot \mathcal{S}_{\text{Topic}}(x_i)), \tag{7}$$

---

**Algorithm 1** Multi-agent collaborative data selection for LLM pretraining

---

**Require:** Training data $\mathcal{D}$, reference task $\mathcal{D}_{\text{ref}}$, main model $\mathcal{M}$, optimizer $\mathcal{O}$, total training steps $T$, selected
     size $k$, update step $U$, Memory for each agent $\mathcal{H}_{\mathcal{A}}$
 1: Initialize model parameters for main model $\mathcal{M}$
 2: Initialize $\mathcal{D}_k$ as a size-k randomly sampled subset from $D$
 3: **for** $t = 1$ to $T$ **do**
 4:    **if** $t \bmod U = 0$ **then**
 5:        **for** each agent $\mathcal{A}$ **do**
 6:            Sample data points according to agent's predefined sampling distribution
 7:            Compute rewards $\mathcal{R}_{\mathcal{M}}(x_i; \mathcal{D}_{\text{ref}})$ for each sampled data point $x_i$
 8:            **Update agent weight** $\mathbf{w}_{\mathcal{A}} \leftarrow \mathbf{w}_{\mathcal{A}} + \eta_{\mathcal{A}} \cdot \overline{\mathbf{R}}_{\mathcal{A}}$
 9:        **end for**
10:        Compute agent score $\overline{R}_{\mathcal{A}}$ and average score $\overline{R}$ according to Eq. 8
11:        **Update collaborative weight** $\theta_{\mathcal{A}} \leftarrow \theta_{\mathcal{A}} + \eta_{\mathcal{A}} \cdot (\overline{R}_{\mathcal{A}} - \overline{R})$.
12:        Calculate coordinator score $S(x_i)$ for $x_i \in \mathcal{D}$ according to Eq. 7
13:        Select dataset for next training stage $\mathcal{D}_k \leftarrow$ Top-$k(S(x_i))$ for $x_i \in \mathcal{D}$
14:    **end if**
15:    Sample a batch of data $B$ from $\mathcal{D}_k$
16:    **Update Main Model** $\mathcal{M} \leftarrow \mathcal{O}(\mathcal{M}, B)$
17: **end for**

---

where $\mathcal{S}_{\text{Quality}}(x_i)$, $\mathcal{S}_{\text{Domain}}(x_i)$, and $\mathcal{S}_{\text{Topic}}(x_i)$ are the scores calculated by the quality, domain, and topic agents for the sample $x_i$, respectively. And $\theta_{\mathcal{A}} \in \{\theta_{\text{Quality}}, \theta_{\text{Domain}}, \theta_{\text{Topic}}\}$ is the collaborative weight for each agent, which is updated during training process.

**Collaborative weight update.** To dynamically adjust the importance of each agent during various training phases, we modify the agent's collaborative weight based on its overall rewards. We compute the reward of each agent and the average reward across all agents:

$$\overline{R}_{\mathcal{A}} = \frac{1}{|n|} \sum_{j=1}^{n} w_{\mathcal{A}}^{j} \cdot \overline{R}_{\mathcal{A}}^{j}, \quad \overline{R} = \frac{1}{k} \sum \overline{R}_{\mathcal{A}}, \tag{8}$$

This information is then used to update each agent's collaborative weight, which is stored in the agent console's memory for future decision-making:

$$\theta_{\mathcal{A}} \leftarrow \theta_{\mathcal{A}} + \eta_{\mathcal{A}} \cdot (\overline{R}_{\mathcal{A}} - \overline{R}). \tag{9}$$

By continuously refining these weights, the collaboration strategy adapts to optimize overall performance and appropriately adjust the role of each agent throughout different stages of training. The complete training pipeline is outlined in Algorithm 1.

## 4 EXPERIMENTS

We conduct a series of experiments to evaluate the effectiveness of our multi-agent collaborative data selection method. Comprehensively, we find that: (1) In the end-to-end experiments, our approach introduces significant improvement in terms of data efficiency leading to faster convergence for LLM training, and achieves up to $10.5\%$ improvements on average across various language model benchmarks when compared with other baseline approaches (§4.1); (2) We also verify that the design and implementation of the core components in our multi-agent framework design are necessary to reach this advanced performance through a set of carefully designed ablation studies (§4.2).

### 4.1 END-TO-END EXPERIMENTS

We evaluate our multi-agent framework against a wide category of state-of-the-art approaches to compare the data efficiency for LLM pretraining. We train a $1.3$ billion parameter LLAMA-2 architecture model with 30 billion selected tokens.

**Experimental setup.** We first enumerate the experimental setup as below:

- **Pretraining datasets.** We utilize the popular SlimPajama (Soboleva et al., 2023) dataset including 627 billion tokens, which is derived from the RedPajama (Computer, 2023) dataset. The

Table 1: Our approach improves model performance across multiple tasks. To ensure all demonstrations fit within the 1024-token context window, we present a comprehensive results table for 0-shot, 3-shot, and 5-shot scenarios in Appendix A.8. The selected tasks in this table include: Problem Solving: ARC-Easy (3), ARC-Challenge (3), MathQA (3), MMLU (3); Commonsense Reasoning: OpenBookQA (3), SocialIQA (3), Winogrande (0), CommonsenseQA (5); and Reading Comprehension: BoolQ (0), Race (5). For the QuRating and DSIR methods, we use their best-performing variants: QuRating-Edu and DSIR-Wiki, respectively. Accuracy is reported with the highest value in each column shown in **bold**.

| Selection Method | Problem Solving (4 tasks) | Commonsense Reasoning (4 tasks) | Reading Comprehension (2 tasks) | Average (10 tasks) |
|---|---|---|---|---|
| **Random sampling** - 30B tokens | 31.1 | 32.9 | 43.1 | 34.2 |
| **Random sampling** - 60B tokens | $33.6^{\uparrow 2.5}$ | $33.7^{\uparrow 0.8}$ | $46.1^{\uparrow 3.0}$ | $36.1^{\uparrow 1.9}$ |
| **Perplexity** PPL (Ankner et al., 2024) | $29.9^{\downarrow 1.2}$ | $30.5^{\downarrow 2.4}$ | $42.4^{\downarrow 0.7}$ | $32.7^{\downarrow 1.5}$ |
| **Classifier-based data selection** | | | | |
| QuRating (Wettig et al., 2024) | $34.1^{\uparrow 3.0}$ | $34.1^{\uparrow 1.2}$ | $41.4^{\downarrow 1.7}$ | $35.6^{\uparrow 1.4}$ |
| FineWeb-Edu (Penedo et al., 2024) | $32.6^{\uparrow 1.5}$ | $33.0^{\uparrow 0.1}$ | $45.3^{\uparrow 2.2}$ | $35.3^{\uparrow 1.1}$ |
| DSIR (Xie et al., 2023b) | $30.9^{\downarrow 0.2}$ | $32.0^{\downarrow 0.8}$ | $41.5^{\downarrow 1.6}$ | $33.5^{\downarrow 0.7}$ |
| **Domain mixing methods** | | | | |
| DOGE Fan et al. (2024) | $30.9^{\downarrow 0.2}$ | $32.2^{\downarrow 0.7}$ | $45.1^{\uparrow 2.0}$ | $34.3^{\uparrow 0.1}$ |
| DoReMi (Xie et al., 2023a) | $30.4^{\downarrow 0.7}$ | $32.6^{\downarrow 0.3}$ | $44.8^{\uparrow 1.7}$ | $34.1^{\downarrow 0.1}$ |
| DMLaw (Ye et al., 2024) | $30.2^{\downarrow 0.9}$ | $32.1^{\downarrow 0.9}$ | $45.1^{\uparrow 2.0}$ | $33.9^{\downarrow 0.3}$ |
| RegMix (Liu et al., 2024) | $30.7^{\downarrow 0.4}$ | $32.5^{\downarrow 0.4}$ | $44.6^{\uparrow 1.5}$ | $34.2^{\uparrow 0.0}$ |
| **Influence** MATES (Yu et al., 2024) | $30.9^{\downarrow 0.2}$ | $34.0^{\uparrow 1.1}$ | $\mathbf{46.5}^{\uparrow 3.4}$ | $35.3^{\uparrow 1.1}$ |
| **Multi-agent collaboration (ours)** | $\mathbf{36.7}^{\uparrow 5.6}$ | $\mathbf{34.8}^{\uparrow 1.9}$ | $45.9^{\uparrow 2.8}$ | $\mathbf{37.8}^{\uparrow 3.6}$ |

SlimPajama (Soboleva et al., 2023) provide the meta-data about the domain information for each sample. Before the training process, we annotate the entire dataset using the FineWeb-Edu quality scorer (Penedo et al., 2024) along with our custom-trained BERT-based topic classifier. The training details for the topic classifier is provided in Appendix §A.1.

- **Training details.** We adopt the model architecture from LLAMA-2 (Touvron et al., 2023b) at the scale of 1.3 billion parameters (see the detailed configuration in Appendix §A.7-Table 8). Following the principles of the scaling law (Hoffmann et al., 2022) and the DCLM framework (Li et al., 2024), we decide to use a total of 30 billion tokens. All training tokens are sampled from the 670 billion-token SlimPajama (Soboleva et al., 2023) dataset using various sampling strategies. Further details regarding the training process can be found in §A.7.

- **Evaluation benchmarks.** To evaluate the pre-trained models thoroughly, we conduct extensive assessments across various downstream tasks, categorized into three areas: (1) problem solving: MMLU (Hendrycks et al., 2021), ARC-Easy/Challenge (Clark et al., 2018), and MathQA Welbl et al. (2017); (2) commonsense reasoning: SIQA (Sap et al., 2019), WinoGrande (Sakaguchi et al., 2020), OpenbookQA (Mihaylov et al., 2018), and CommonsenseQA (Talmor et al., 2019); (3) reading comprehension: RACE (Lai et al., 2017) and BoolQ (Clark et al., 2019). Evaluations are conducted using the `lm-evaluation-harness` framework (Gao et al., 2023) in an in-context learning setting, and average accuracy is reported for easy comparison.

- **Baselines.** We select a wide range of baselines to conduct extensive the data efficiency comparison, where these methods can be classified to five main categories: (1) *random sampling*, we test this policy with both the standard data volume of 30B tokens and a supplemented version with 60B tokens; (2) *perplexity*-based data selection Ankner et al. (2024); (3) *classifier*-based data selection, where we select the following methods: QuRating Wettig et al. (2024), FineWeb-Edu Penedo et al. (2024), DSIR-Book Xie et al. (2023b) and DSIR-Wiki Xie et al. (2023b); (4) *domain mixing*-based methods, where we select the following methods: DOGE Fan et al. (2024), DoReMi Xie et al. (2023a), DMLaw Ye et al. (2024) and RegMix Liu et al. (2024); and (5) *influence function* based methods for online data selection, i.e., MATES Yu et al. (2024). Implementation details of these baselines can be found in §A.7.2.

**Results.** We present the results of three types of downstream tasks in Table 1, with the complete 0-shot (Table 10), 3-shot (Table 11), and 5-shot (Table 12) results for all tasks enumerated in §A.8. We highlight that *our methods show a substantial improvement in the average performance across all downstream tasks when compared with all the baselines.* Concretely, we observe that when compared with the *random sampling* based approach, our method not only significantly outperforms the standard 30 billion token setup but also surpasses the model trained on 60 billion tokens with a performance gain of 4.7%. Similarly, we also show an improvement of 15.6% compared with *perplexity*-based data selection Ankner et al. (2024), an improvement of up to 6.2% compared with *classifier*-based data selection, an improvement of up to 10.2% compared with *domain mixing*-based methods, and an improvement of 7.1% compared with *influence function* based approach, i.e., MATES Yu et al. (2024).

**Discussion.** We highlight that our proposed multi-agent collaborative data selection mechanism introduces statistical efficiency in terms of LLM training convergence and also provides some computational efficiency in terms of data processing overheads. In terms of *statistical efficiency*, our method consistently outperforms others at every benchmarked training step, as shown in Figure 4. While MATES Yu et al. (2024) performs comparably to our methods during the early training phase (steps 1500 to 3000), its performance drops in later stages. This aligns with its original paper, which notes that relying solely on influence functions for specific reference tasks (e.g., LAMBADA (Paperno et al., 2016)) can degrade performance in mid-to-late pretraining. Despite this, MATES still outperforms other methods without dynamic adjustments shown in Figure 4. In contrast, our multi-agent collaborative data selection mechanism can dynamically adjust the corresponding weights from different agents and select data based on the most up-to-date model preferences, effectively mitigating biases and surpassing other domain-mixing and data-selection techniques. In terms of *computational efficiency*, we also achieve higher computational efficiency than previous methods. For example, QuRating Wettig et al. (2024) requires around $7.13 \times 10^{20}$ FLOPs to label the entire SlimPajama dataset, while our offline labeling takes just $9.91 \times 10^{19}$ FLOPs. MATES Yu et al. (2024), which recalculates influence scores and trains a BERT model for each labeling cycle, incurs $1.98 \times 10^{20}$ FLOPs for a four-stage update. Additionally, MATES' labels are only usable in the next training stage, making it time-consuming and difficult to scale. In contrast, our method can improve the computational efficiency from two aspects: (1) we find that a group of light-weight agents collaboratively enables superior data selection, which is more computational efficiency than any method that requires a heavy data processing or label procedure; (2) the collaborative, dynamic learning procedure introduced in our multi-agent framework is computational efficient; by using a sampled holdout set and CPU-based calculations for updating agent parameters, our computational overhead is ignorable compared with heavy LLM training computation.

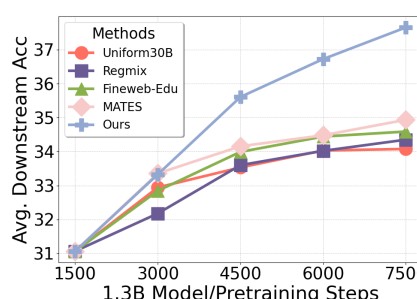

Figure 4: Downstream three-shot performance of the 1.3B model in relation to pretraining steps, using 7500 steps for 30B tokens. Our methods outperform baselines from all categories.

### 4.2 ABLATION STUDY

We introduce a set of carefully designed ablation studies to justify the design and implementation of our multi-agent collaborative data selection framework. Concretely, (1) we test the combination of different agents to show the advance introduced by collaboration, and (2) we verify the necessity of the dynamic adjustment of the agent's weight for data selection.

**Results and discussion.** The results of the ablation study are shown in Table 2. We want to highlight the result from two aspects: First, the ablation study underscores *the importance of each agent in achieving optimal performance* across the training tasks. When the quality, domain, and topic agents are used together, the model performs best, highlighting the benefits of their combined use, as shown in Table 2. In terms of evaluating *each agents' contributions*, we find that the quality agent excels in problem-solving tasks like ARC-E and MathQA by leveraging educational knowledge but is less effective for domain-specific or context-heavy tasks like BoolQ and RACE. The domain agent enhances commonsense reasoning (e.g., CommonsenseQA) and reading comprehension (e.g., BoolQ) by incorporating domain-specific knowledge. The topic agent is most effective for multi-topic tasks

Table 2: This ablation study examines the performance of various combinations of agent collaboration and update mechanisms. All models are in 1.3B LLaMA2 architecture. Three-shot accuracy is reported for all tasks, with the highest value in each column shown in **bold**.

| Selection Method | Problem Solving | | | | Commonsense Reasoning | | | | Reading Compreh. | | |
|---|---|---|---|---|---|---|---|---|---|---|---|
| | **ARC-E** | **ARC-C** | **MathQA** | **MMLU** | **O.B.QA** | **SIQA** | **W.G.** | **C.S.QA** | **BoolQ** | **RACE** | **Average** |
| Quality&Domain&Topic Agent | **65.8** | **31.5** | **23.0** | 26.6 | **24.6** | 39.9 | **54.1** | 20.1 | **60.4** | 30.5 | **37.7** |
| without collaboration update | 59.4 | 26.3 | 21.3 | 25.1 | 20.5 | 38.9 | 52.9 | 19.8 | 58.1 | 28.3 | 35.1 |
| Domain&Quality Agent | 63.3 | 29.7 | 22.6 | 25.1 | 21.8 | **40.5** | 53.1 | 20.3 | 59.5 | 28.8 | 36.5 |
| Topic&Quality Agent | 62.9 | 28.1 | 22.3 | 26.5 | 22.6 | 39.6 | 51.8 | **21.7** | 56.7 | **30.7** | 36.3 |
| Domain&Topic Agent | 55.6 | 25.2 | 21.8 | 26.5 | 23.1 | 39.1 | 53.7 | 20.9 | 57.5 | 29.0 | 35.2 |
| Quality Agent | 59.1 | 29.7 | 22.4 | 25.3 | 21.1 | 38.5 | 51.2 | 19.1 | 57.2 | 28.3 | 35.2 |
| Domain Agent | 54.1 | 25.6 | 21.4 | 25.9 | 22.3 | 38.1 | 53.6 | 20.0 | 58.1 | 27.9 | 34.7 |
| Topic Agent | 55.3 | 25.3 | 21.9 | **27.1** | 22.1 | 39.4 | 51.5 | 19.8 | 56.3 | 28.9 | 34.8 |
| No Agent | 54.6 | 23.0 | 22.1 | 24.9 | 18.8 | 40.3 | 52.9 | 21.5 | 53.0 | 29.8 | 34.1 |

like MMLU and contributes significantly to commonsense reasoning tasks like SocialIQA. Second, the ablation study verifies the design and implementation of *the collaborative dynamic adjustment of the agents' weights* (introduced §3.4) for efficient data selection. When agents were initialized with equal, fixed weights instead of using dynamic weighting, overall performance dropped significantly, as shown in Table 2. This highlights the critical role of dynamic weight adjustment in the collaborative framework. Detailed analysis is in Appendix A.5.1.

## 5 RELATED WORK

**Data selection in LLM pretraining.** Selecting high-quality pretraining data from large corpora is crucial for effective LLM training. Recent approaches leverage various methodologies for efficient data selection. Concretely, *classifiers* (Brown, 2020; Chowdhery et al., 2023; Du et al., 2022; Xie et al., 2023b) and *language modeling perplexity* (Wenzek et al., 2020; Thrush et al., 2024) have been applied to identify data resembling high-quality samples; recently, more advanced quality scores based on classifier have shown the effectiveness in data selection, e.g., QuRating (Wettig et al., 2024), FineWeb-Edu (Lozhkov et al., 2024), etc. *Data mixture* is another effective way to improve data diversity, at both token level (Touvron et al., 2023a; Gao et al., 2020; Soboleva et al., 2023) and sample level, e.g., DoReMi (Xie et al., 2023a), DOGE (Fan et al., 2024), DMLaw (Ye et al., 2024), and RegMix (Liu et al., 2024); very recently, topic distributions has also been considered as an effective data mixing method, e.g., the downsampling overrepresented topics in Llama 3.1 (Team, 2024). *Influence functions* have been studied to understand for data efficiency (Koh & Liang, 2017), and some recent attempts based on efficient approximation have been proposed to improve data efficiency in LLM pretraining (Schioppa et al., 2022; Grosse et al., 2023; Isonuma & Titov, 2024); for example, MATES (Yu et al., 2024) uses a staged BERT model to assess data influence, QUAD (Zhang et al., 2024) leverage cluster information to reduce the computational cost of calculating individual data influence.

**Multi-agent collaborative frameworks.** Multi-agent collaborative frameworks (Russell & Norvig, 2016; Wooldridge, 2009) facilitate cooperative problem solving among autonomous agents and have been widely applied to solve various problems, e.g., neural optimizer search (Bello et al., 2017), collaborative LLM programming (Hong et al.). In these systems, agents may have conflicting goals and independently take actions based on their objectives; a *reward mechanism* evaluates these actions, providing feedback that allows each agent to adjust and refine its strategy over time; a *central console* coordinates the agents by synthesizing their feedback and guiding the overall system towards more optimal decisions (Russell & Norvig, 2016; Wooldridge, 2009). Collaboration among agents is dynamic, as they adapt their ability to work together improves (Olfati-Saber, 2006).

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

# A APPENDIX

## A.1 DETAILS OF TRAINING TOPIC CLASSIFIER

As shown in Figure 3, we first cluster 1.4 billion documents obtained from Common Crawl (Project, 2007) into 10,000 clusters using KNN. And we use GPT-4o (OpenAI, 2024) to generate a summary for the content in each cluster. Additionally, we implement two parallel steps: unsupervised and supervised. In the unsupervised step, we perform secondary clustering of the 10,000 clusters into 100 clusters, from which we extract 20 summaries for each cluster. We utilize GPT-4o to extract category labels, refining these into a coherent hierarchical labeling system for the classification of 42 distinct topics.

In the supervised data processing step, leveraging Gopher cleaning ruls (Rae et al., 2021) and Min-Hash (Broder, 1997) deduplication, we clean the whole datasets and cluster the datasets into 10,000 clusters. We then extract 50 equidistant samples from each cluster. This process yields approximately 500,000 data points, which we categorize into the aforementioned 42 topics by calling GPT-4o (OpenAI, 2024) using the prompt shown below:

> **Prompt Template**
>
> **Task:** You are an annotator responsible for analyzing the category distribution of web data. For each provided full-text segment, select one topic from the following list of 42 that best represents the primary focus of the text:
>
> **Topic:** [Animals, Culture, Travel, Politics, Art, Architecture, Consumerism, Social Networking, Investment, Environment, Industry, Technology, Entertainment, Hobbies, Music, History, Healthcare, Fashion, Beauty, Relationships, Real Estate, Careers, Agriculture, Local, Harmful information, Adult, Sports, News, Pets, Community, Home, Health, Business, Literature, Food Culture, Transportation, Law, Science and Technology, Finance, Education, Automotive, Society]
>
> **Response Format:** Please respond with a JSON object structured as follows:
> {
> "selected topic": string //selected topic from the 42 topics
>  "explanation": string //explain why reorganize the topic in this way
> }

Figure 5: We illustrate the prompt construction process for GPT-4 to reorganize the topic of 500k data points.

Since GPT-4o is not specialized for classification tasks, we obtain actual topic data with slightly more than 42 topics, as shown in Figure 6. We then manually summarize the topics provided by GPT-4 into 13 categories, ensuring that the subtopics within each category shared similarities. The detailed category distributions appear in Figure 6, along with specific clustering information. Ultimately, we employ the annotated data to fine-tune a BERT-like regression model (Devlin et al., 2018). Following model classification, we conduct human proofreading to ensure accuracy, and we present the final results below.

## A.2 GUIDELINES FOR GENERALIZING A NEW CRITERIA AS AN AGENT

This section provides a detailed guidelines for incorporating a new criterion into our multi-agent system. The process is designed to ensure seamless integration and effective collaboration between existing and new agents.

Our framework offers several significant benefits when integrating a new agent. First, it offers **flexibility**, as the addition of new criteria can be performed independently of the core framework. This decoupling ensures that introducing new components does not require significant structural changes, allowing for smooth integration with minimal disruption to existing processes. Second, the approach is highly **scalable**. By enabling the training of new classifiers offline, the system

---

**Algorithm 2** Integrating a New Criterion into Multi-Agent Collaboration

---

**Require:** Sampled dataset $\mathcal{D}_{\text{sample}}$, pretraining dataset $\mathcal{D}_{\text{train}}$, reference dataset $D_{\text{ref}}$, existing agents $\{\mathcal{A}_i\}$, scoring weights $\{\theta_i\}$, memory $\mathcal{H}_{\mathcal{A}}$ for each agent.
1: **Annotating Data for the New Criteria**
2:     Sample data from the whole datasets, and annotate sampled dataset $\mathcal{D}_{\text{sample}}$ according to new criterion.
3: **Training a Classifier for the New Criteria**
4:     Train a supervised classifier on $\mathcal{D}_{\text{sample}}$.
5: **Defining the New Agent**
    • **Action Space:** Sample and assess data points across subcategories of the new Criteria.
    • **Memory:** Store prior scores and update based on model feedbacks.
6: **Labeling the Pretraining Dataset**
7:     Use trained classifier to label the whole training dataset $\mathcal{D}_{\text{train}}$ and store these labels in memory $\mathcal{H}_{\mathcal{A}_{\text{new}}}$.
8: **Defining Agent Weights and Collaboration Strategy**
    1. Define the subcategory weights updating mechanism for $\mathcal{A}_{\text{new}}$ using Eq. 5:

$$w_{\text{new}}^j \leftarrow (1 - \eta_{\text{new}}) \cdot w_{\text{new}}^j + \eta_{\text{new}} \cdot R_{\text{new}}^j.$$

    2. Integrate $\mathcal{A}_{\text{new}}$ into the scoring function using Eq. 7:

$$S(x_i) = \theta_{\text{Quality}} S_{\text{Quality}}(x_i) + \theta_{\text{Domain}} S_{\text{Domain}}(x_i) + \theta_{\text{Topic}} S_{\text{Topic}}(x_i) + \theta_{\text{new}} S_{\text{new}}(x_i).$$

9: **Agent Initialization with Regression Techniques**
10:     Initialize the agent weights $w_{\text{new}}$ using regression techniques (e.g., `RegMix`).

---

can be easily adapted to handle a wide variety of data selection goals, which can evolve over time as new criteria emerge. Finally, the framework ensures **extensibility**, meaning it can seamlessly accommodate new objectives, whether simple or complex. This extensibility is key to maintaining efficient and effective collaboration across multiple agents, regardless of the size or complexity of the task at hand. As demonstrated in Algorithm 1, our framework seamlessly integrates a new agent through a series of straightforward steps. This approach ensures that the multi-agent framework remains flexible and effective in addressing diverse data selection objectives while preserving its collaborative efficiency.

### A.3 CONNECTION OF MULTI-AGENT COLLABORATIVE SELECTION METHOD TO MULTI-AGENT RL

The proposed multi-agent collaborative selection method is fundamentally inspired by the intelligent agent defined in Russell & Norvig (2016), where the agent generally refers to an entity that perceives some status and map the observed status into actions. However, our framework also has many similarity compare with traditional multi-agent framework in reinforcement learning, where multiple agents work together to optimize a shared objective. In this section, we formally demonstrate the relationship between our framework and traditional Multi-agent RL.

#### A.3.1 OVERALL DEFINITION IN REINFORCEMENT LEARNING FORMULATION

We first clearly formulate each component of framework compare with components in general MARL framework for understanding the mechanism of our framework. As our goal is to select the opt the **global action** at each step involves selecting a subset of data, $\mathcal{D}_k$, from the entire dataset, $\mathcal{D}$. This subset is used to update the model, where batches are drawn from $\mathcal{D}_k$ for training. The **global state** is represented by the current model parameters, $M$, which evolve as the model is trained. The **state transition** is formalized as:

$$M' = \mathcal{O}(M, \mathcal{D}_k), \tag{10}$$

where $M'$ denotes the updated model after training on $\mathcal{D}_k$.

The **reward function** measures the improvement in model performance on a reference task, $\mathcal{T}_{\text{ref}}$, which serves as a proxy for the true downstream task $\mathcal{T}_{\text{eval}}$. The reward is defined as:

$$R(M'|M, \mathcal{T}_{\text{ref}}) = \mathbb{E}_{x \sim \mathcal{T}_{\text{ref}}}[-l(x; M')], \tag{11}$$

where $l(x; M')$ is the loss on $\mathcal{T}_{\text{ref}}$. For individual data points, the reward is estimated using influence functions:

$$\mathcal{R}(x_i | M, \mathcal{T}_{\text{ref}}) = \text{Influence}(x_i, M, \mathcal{T}_{\text{ref}}). \tag{12}$$

This formulation links data selection directly to its impact on improving the model's performance on $\mathcal{T}_{\text{ref}}$.

### A.3.2 AGENT DESIGN

The estimation of value is as follows: The agent stores a reward estimation vector for each subset. The update rule is given by

$$w_{\mathcal{A}}^j(t+1) = (1 - \eta_{\mathcal{A}}) \cdot w_{\mathcal{A}}^j(t) + \eta_{\mathcal{A}} \cdot \bar{R}_{\mathcal{A}}^j. \tag{13}$$

The sliding average is used here because if all data in a subset were fully processed to compute $\bar{R}_{\mathcal{A}}^j$, there would be no need for a sliding average. However, since only a portion of the data is sampled, the estimate has higher variance, which is not favorable for training. At the same time, the influence score itself is dynamic (even if the data remains constant, the model evolves). Averaging with outdated scores introduces bias. Therefore, the sliding average factor $\eta_{\mathcal{A}}$ strikes a 'bias-variance tradeoff'. We assume the score estimate for each data point $x_i$ in $\mathcal{D}_{\mathcal{A}}^j$ with respect to the dimension of interest for the agent, is given by

$$S_{\mathcal{A}}(x_i) = w_{\mathcal{A}}^j, \quad \text{where} x_i \in \mathcal{D}_{\mathcal{A}}^j. \tag{14}$$

### A.3.3 MULTI-AGENT COLLABORATION

Assume that the score of a single data point in the reference task is obtained as a weighted sum of multiple components. The total score for each data point is given by Equation 8 as:

$$S(x_i) = \sum_{\mathcal{A} \in \text{set}(\mathcal{A})} \theta_{\mathcal{A}} \cdot S_{\mathcal{A}}(x_i), \tag{15}$$

where $\theta_{\mathcal{A}}$ are collaborative weights. A central coordinator adjusts these weights over time based on the agents' contributions to the overall reward:

$$\theta_{\mathcal{A}}(t+1) = \theta_{\mathcal{A}}(t) + \eta_{\mathcal{A}}(\bar{R}_{\mathcal{A}} - \bar{R}), \tag{16}$$

where $\bar{R}_{\mathcal{A}}$ is the agent's average reward, and $\bar{R}$ is the global average reward:

$$\bar{R}_{\mathcal{A}} = \frac{1}{n} \sum_{j=1}^n w_{\mathcal{A}}^j \cdot \bar{R}_{\mathcal{A}}^j, \quad \bar{R} = \frac{1}{|\text{set}(\mathcal{A})|} \sum_{\mathcal{A} \in \text{set}(\mathcal{A})} \bar{R}_{\mathcal{A}}. \tag{17}$$

We consider three possible cases for our framework, comparing its relationship with traditional optimization problem.

- **Single-agent case**: If only one agent is involved, $\theta$ becomes irrelevant, reducing the problem to a classical optimization scenario where the agent greedily selects the optimal data based on one criteria.

- **Multi-agent competitive mechanism**: When multiple agents are present, $\theta$ reflects each agent's capability. Selecting the best-performing agent for decision-making introduces a heuristic competitive mechanism, building upon the classical optimization framework.

- **Multi-agent collaborative mechanism**: Alternatively, when multiple agents are involved, $\theta$ can be used to weigh each agent's contributions for decision-making. This introduces a smoother heuristic cooperative mechanism, extending the classical optimization framework by leveraging weighted collaboration. This heuristic cooperative mechanism dynamically adjusts the influence of each agent based on the model's current preferences, enabling more effective data filtering decisions.

In practice, we choose to use the multi-agent collaborative mechanism for data selection. We have added comparisons with single-agent and competitive mechanisms in Table 3 to further elaborate the effectiveness of collaboration.

Table 3: This ablation study examines the performance of various combinations of agent collaboration (**Agent**) and dynamic collaborative weight update (**Dynamic**). Accuracy is reported for all tasks, with the highest value in each column shown in **bold**.

| | | Problem Solving | | | | Commonsense Reasoning | | | | Reading Compreh. | | |
|---|---|---|---|---|---|---|---|---|---|---|---|---|
| **Agent** | **Dynamic** | **ARC-E** | **ARC-C** | **MathQA** | **MMLU** | **O.B.QA** | **SIQA** | **W.G.** | **C.S.QA** | **BoolQ** | **RACE** | **Average** |
| with | with | **65.8** | **31.5** | **23.0** | **26.6** | **24.6** | **39.9** | **54.1** | 20.1 | **60.4** | **30.5** | **37.7** |
| with | without | 59.4 | 26.3 | 21.3 | 25.1 | 20.5 | 38.9 | 52.9 | 19.8 | 58.1 | 28.3 | 35.1 |
| without | - | 59.2 | 26.1 | 20.3 | 25.4 | 21.3 | 39.1 | 52.6 | **20.1** | 56.5 | 29.1 | 35.0 |

## A.4 DETAILED ANALYSIS OF COMPUTATIONAL OVERHEAD

In this subsection, we compare the computational overhead of our multi-agent collaboration framework with baseline approaches. The analysis focuses on three aspects: **offline computation efficiency**, **online computation efficiency**, and **overall FLOPs requirements**. Table 4 summarizes these comparisons.

Table 4: Comparison of Computational Overhead Across Methods

| **Selection Method** | **Offline Computation Cost (FLOPs)** | **Online Computation Cost (FLOPs)** | **Overall (FLOPs)** |
|---|---|---|---|
| Qu-Rating (Wettig et al., 2024) | $7.13 \times 10^{20}$ | N.A. | $7.13 \times 10^{20}$ |
| MATES (Yu et al., 2024) | N.A. | $1.99 \times 10^{20}$ | $1.99 \times 10^{20}$ |
| Multi-agent collaboration (ours) | $9.91 \times 10^{19}$ | $1.19 \times 10^{18}$ | $1.00 \times 10^{20}$ |

### A.4.1 OFFLINE COMPUTATION EFFICIENCY

Our method achieves superior offline efficiency by requiring only $9.91 \times 10^{19}$ FLOPs for a one-time dataset labeling process using a 109M BERT-based model for inference. This is nearly an order of magnitude more efficient than Qu-Rating, which consumes $7.13 \times 10^{20}$ FLOPs due to its reliance on a larger 1.3B Sheared-LLaMA model. MATES does not utilize offline computation, relying solely on online updates, which avoids this cost but limits its flexibility and scalability. The offline labeling in our method ensures robust initial scores for large-scale datasets while laying the groundwork for efficient online updates.

### A.4.2 ONLINE COMPUTATION EFFICIENCY

For adaptive online updates, both our approach and MATES compute influence scores with $1.19 \times 10^{18}$ FLOPs. However, MATES involves labeling the entire dataset with a 109M BERT-based model in every round, amounting to $1.98 \times 10^{20}$ FLOPs across four data selection stages. In contrast, our method avoids re-labeling the entire dataset, significantly reducing the computational cost by focusing on labeling the large pretraining datasets only once.

Overall, our approach cuts the computational cost in half compared to MATES and requires only about 1/7 of the computational resources used by Qu-Rating.

## A.5 ANALYSIS OF ABLATION STUDY

### A.5.1 ANALYSIS OF AGENT ROLES ON DIFFERENT TYPE OF TASKS

We show the agent ablation study conducted on 373M LLaMA2 models Table 5 as well as 1.3B LLaMA2 models Table 2. First, the ablation study underscores *the importance of each agent in achieving optimal performance* across the training tasks. When the quality, domain, and topic agents are used together, the model performs best, highlighting the benefits of their combined use,

Table 5: This ablation study examines the performance of various combinations of agent collaboration and update mechanisms. All models are in 373M LLaMA2 architecture. Accuracy is reported for all tasks, with the highest value in each column shown in **bold**.

| Selection Method | Problem Solving | | | | Commonsense Reasoning | | | | Reading Compreh. | | |
|---|---|---|---|---|---|---|---|---|---|---|---|
| | ARC-E | ARC-C | MathQA | MMLU | O.B.QA | SIQA | W.G. | C.S.QA | BoolQ | RACE | Average |
| Quality&Domain&Topic Agent | **57.9** | **24.7** | **21.9** | 25.4 | **20.2** | **37.9** | **52.6** | **20.4** | 59.6 | **29.4** | **35.0** |
| without collaboration update | 47.9 | 20.4 | 21.0 | 25.1 | 17.2 | 37.3 | 51.3 | 20.0 | 56.5 | 28.3 | 32.5 |
| Domain&Quality Agent | 55.1 | 18.6 | 21.7 | 24.4 | 17.4 | 37.1 | 51.2 | 19.8 | **61.7** | 28.2 | 33.5 |
| Topic&Quality Agent | 56.2 | 24.4 | 21.8 | 25.2 | 19.4 | 36.3 | 49.0 | 19.7 | 56.1 | 28.5 | 33.6 |
| Domain&Topic Agent | 44.6 | 18.3 | 21.7 | 25.7 | 16.2 | 36.6 | 51.9 | 19.9 | 61.6 | 27.8 | 32.4 |
| Quality Agent | 53.0 | 24.7 | 21.8 | 25.5 | 18.0 | 36.3 | 49.5 | 18.1 | 57.0 | 28.0 | 32.9 |
| Domain Agent | 44.1 | 19.1 | 20.8 | 25.6 | 16.6 | 36.8 | 52.0 | 19.7 | 56.7 | 28.2 | 32.0 |
| Topic Agent | 42.7 | 19.2 | 21.0 | **27.0** | 17.4 | 37.1 | 50.7 | 19.7 | 54.6 | 28.5 | 31.8 |
| No Agent | 42.5 | 20.0 | 21.1 | 23.8 | 14.6 | 35.9 | 50.1 | 18.8 | 56.1 | 27.9 | 31.1 |

Table 6: This ablation study examines the performance of various combinations of reference task. Accuracy is reported for all tasks, with the highest value in each column shown in **bold**.

| Reference Tasks | Problem Solving | | | | Commonsense Reasoning | | | | Reading Compreh. | | |
|---|---|---|---|---|---|---|---|---|---|---|---|
| | ARC-E | ARC-C | MathQA | MMLU | O.B.QA | SIQA | W.G. | C.S.QA | BoolQ | RACE | Average |
| LAMBADA&SQuAD&Jeopardy | **65.8** | **31.5** | 23.0 | 26.6 | 24.6 | 39.9 | 54.1 | 20.1 | **60.4** | **30.5** | **37.7** |
| LAMBADA | 64.3 | 31.2 | 22.3 | **26.8** | 23.5 | 39.6 | **54.6** | 20.4 | 59.6 | 30.1 | 37.2 |
| SQuAD | 65.1 | 30.9 | **23.4** | 25.9 | **24.9** | 40.1 | 53.8 | 21.2 | 59.1 | 29.3 | 37.4 |
| Jeopardy | 63.9 | 30.3 | 23.6 | 26.3 | 24.1 | **40.7** | 54.5 | **21.8** | 59.1 | 30.2 | 37.5 |
| Random selection | 54.6 | 23.0 | 22.1 | 24.9 | 18.8 | 40.3 | 52.9 | 21.5 | 53.0 | 29.8 | 34.1 |

as shown in Table 2. In terms of evaluating *the performance of each agent*, we find that the quality agent outperforms other single-agent configurations, excelling in problem solving tasks. However, its performance drops on tasks requiring domain knowledge or contextual understanding. Here, the domain and topic agents play a crucial role, as they excel in these areas. Despite this, neither performs well on problem solving tasks, except for the topic agent, which significantly improves MMLU performance, indicating that topic diversity may benefit such tasks. In terms of evaluating *the combination of the agents*, we find that removing any agent noticeably reduces overall accuracy, though the impact varies. Excluding the quality agent leads to the largest drop, significantly affecting performance in problem solving tasks, and commonsense reasoning tasks like OpenbookQA. This highlights the quality agent's vital role in reasoning and problem-solving. Similarly, excluding the topic agent causes a performance drop in ARC-Challenge and a significant reduction in MMLU, emphasizing its importance in tasks covering diverse subjects; removing the domain agent results in a performance drop in commonsense reasoning tasks, underscoring its key contribution to these areas. Second, the ablation study verifies the design and implementation of *the collaborative dynamic adjustment of the agents' weights* (introduced §3.4) for efficient data selection. Concretely, in this variant, all agents were initialized with equal weights, which remained fixed throughout training without adjusting for individual agent performance. Surprisingly, this fixed-weight approach (equal to random sampling) resulted in a significant drop in overall performance compared to the dynamic weighting used in the collaborative update framework, as shown in Table 2. We believe this result from the ablation study is a strong indicator that the dynamic adjustment of the celebration mechanism is essential for efficient data selection.

## A.5.2 ABLATION STUDY ON REFERENCE TASKS SELECTION

In our experiments of selecting reference tasks in Table 6, we observe that while the choice of reference tasks can influence performance, the impact on average performance is marginal (within 0.5 points). Using different reference tasks consistently leads to a significant improvement in average performance compared to random data selection, demonstrating that our method is not sensitive to the choice of reference tasks.

Table 7: This ablation study compares the performance of a **3B model** using random sampling versus multi-agent collaboration. Accuracy is reported for all tasks, with the highest value in each column shown in **bold**.

| Selection Method | Problem Solving | | | | Commonsense Reasoning | | | | Reading Compreh. | | |
|---|---|---|---|---|---|---|---|---|---|---|---|
| | ARC-E | ARC-C | MathQA | MMLU | O.B.QA | SIQA | W.G. | C.S.QA | BoolQ | RACE | Average |
| Random Sampling | 34.8 | 17.7 | 21.3 | 23.0 | 12.0 | 32.9 | 50.2 | 19.6 | 37.8 | 20.9 | 27.0 |
| Multi-agent collaboration (Ours) | **42.9** | **21.3** | **21.9** | **24.0** | **15.8** | **33.9** | **51.0** | **20.4** | **54.8** | **21.2** | **30.7** |

## A.6 GENERALIZATION TO 3.6B MODELS

To evaluate the scalability of our approach, we conducted additional experiments training a 3.6 billion parameter model based on the LLaMA 3.2 architecture, which further demonstrates the scalability of our method. So far we have trained on 36 billion tokens and achieved strong performance, with plans to continue training with additional tokens according to scaling laws. As Table 7 shows, when compared to random selection, our method shows consistent performance improvements across all downstream tasks, achieving a 13.7% increase in average accuracy—significantly higher than the 10.5% improvement observed with the 1.3B models. Based on trends across three different model sizes (373M, 1.3B, and 3.6B), our approach consistently outperforms random selection by over 10% on average. This consistent advantage makes us believe that it suggests our method has strong potential for training even larger models, including those with 10B+ parameters

## A.7 DETAILS OF PRETRAINING

### A.7.1 DETAILS OF PRETRAINING MODELS ARCHITECTURE

The specific architecture of pretraining model is shown in Table 8. Each model was trained on 32x NVIDIA A800, employing a global batch size of $4 \times 2^{20}$ tokens and completing 7,500 steps in about 14 hours. The average token processing rate per GPU was about 20,000 tokens per second. The learning rate was set to $5 \times 10^{-5}$, and the Adam optimizer was employed with hyperparameters ($\beta_1 = 0.9, \beta_2 = 0.95, \epsilon = 10^{-8}$).

Table 8: Architecture of pre-trained decoder-only models.

| Hyperparameter | 370M Model Value | 1.3B Model Value | 3.6B Model Value |
|---|---|---|---|
| Vocabulary Size | 32,000 | 32,000 | 128,256 |
| MLP Ratio | 8/3 | 8/3 | 8/3 |
| Hidden Dimension Size | 2048 | 1024 | 3072 |
| Number of Layers | 24 | 24 | 28 |
| Number of Attention Heads | 16 | 8 | 24 |
| Number of KV Attention Heads | 16 | 8 | 8 |
| RoPE Base | 10,000 | 10,000 | 500,000 |
| Maximum Context Window Length | 1024 | 1024 | 1024 |
| Number of Parameters | 373,867,520 (370M) | 1,345,423,360 (1.3B) | 3,606,752,256 (3.6B) |

### A.7.2 DETAILS OF BASELINE METHOD IMPLEMENTATION

Regarding the classifier methods, QuRating (Wettig et al., 2024) and DSIR (Xie et al., 2023b), we implement QuRating by downloading the open-source checkpoint from Hugging Face. Notably, the released model has a context length of 4096, whereas ours is 1024. However, this discrepancy does not impact our testing tasks, as our maximum of 5-shot examples remains within the 1024 limit. Despite this, we have totally similar model configuration as well as the total number of training tokens with all the checkpoints we downloaded. Similarly, the replication of PPL is based on the

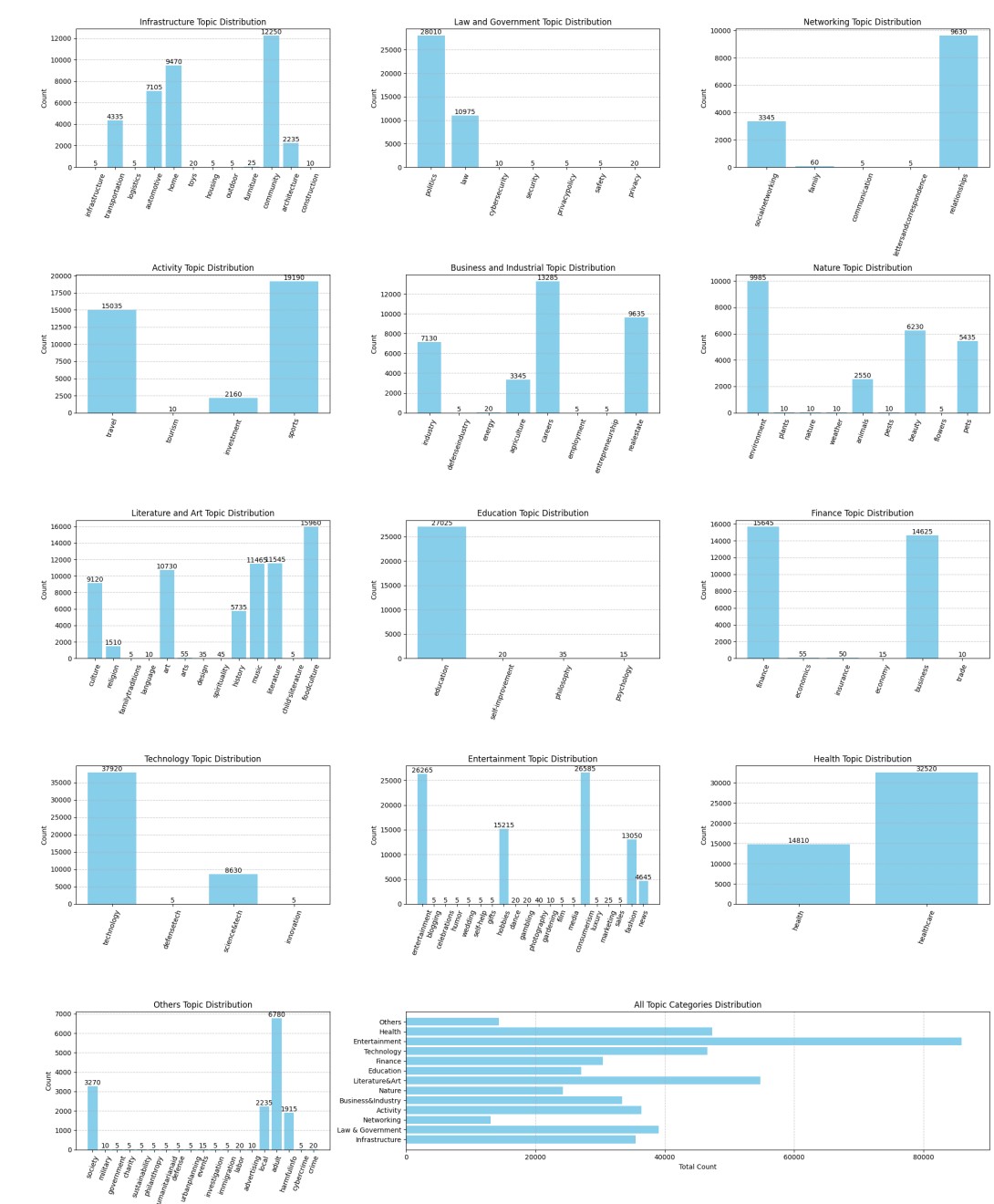

Figure 6: We illustrate the distribution of manually annotated and clustered data, which includes 13 topics: Infrastructure, Law and Government, Networking, Activity, Business and Industry, Nature, Literature and Art, Education, Finance, Technology, Entertainment, Health, and Others.

publicly available checkpoint from the original paper. For FineWeb-Edu (Lozhkov et al., 2024), we download the quality scorer to label all the training data from SlimPajama datasets, and adopt the methodology described in the corresponding publication and train all the model from scratch.

Domain mixing refers to the technique of combining data from different sources or domains to enhance the diversity and robustness of a model's training dataset. In our implementation, we apply various mixing methods: DoReMi (Xie et al., 2023a), DOGE (Fan et al., 2024), DMLaw (Ye et al.,

2024), and RegMix (Liu et al., 2024). Each method contributes distinct proportions of data from specific domains, as reflected in the domain weights presented in Table 9. Notably, the weights indicate the percentage of contributions from each domain.

Table 9: Exact domain weights (%) on SlimPajama obtained by data mixing methods. Abbreviations: C.C. = CommonCrawl, Wiki = Wikipedia, StackEx. = StackExchange

| Mixing Method | C.C. | C4 | GitHub | Books | ArXiv | Wiki | StackEx. |
|---|---|---|---|---|---|---|---|
| SlimPajama | 52.20 | 26.70 | 5.20 | 4.20 | 4.60 | 3.80 | 3.30 |
| DoReMi | 38.11 | 11.41 | 6.54 | 8.19 | 4.24 | 23.07 | 8.47 |
| DOGE | 21.35 | 26.93 | 7.03 | 4.50 | 8.80 | 14.82 | 16.58 |
| DMLaw | 12.50 | 25.00 | 14.06 | 9.38 | 25.00 | 1.56 | 12.50 |
| RegMix | 17.37 | 51.03 | 0.23 | 0.23 | 0.08 | 29.77 | 1.27 |

For the reproduction of MATES (Yu et al., 2024), we start by utilizing Random-Slimpajama at the 1500th training step as our primary pretraining model and fine-tune the BERT-base from the original thesis as our data influence model. During the training of the data influence model, we uniformly sample 1/13 of the data as hold-out data from each area of our dataset and employ LAMBADA (Paperno et al., 2016) as a reference task, following the MATES methodology. Ultimately, we use the trained BERT-base data influence model to predict the entire training dataset, selecting the top 1/20 as our pretraining data. This selection process is executed using the Gumbel-Top-k algorithm (Kim et al., 2016), consistent with MATES. We leverage a four-step updates similar to the original paper, and conduct the above implementation at 1500th, 3000th, 4500th and 6000th model training steps using the current models.

### A.7.3 Details Agent Weight Initialization

We employ an agent weight initialization technique within the RegMix (Liu et al., 2024) framework, which is crucial for the effective training of proxy models. Our dataset is organized into three distinct categories: domain, quality, and topic. For each category, we initialize the data weights based on the original proportions across 512 configurations and subsequently train a TinyLlama-1M with 1 billion tokens as a proxy model for each configuration. We evaluate this model on previously unseen data mixtures, specifically using validation set loss, following RegMix, for assessment. We then fit a regression model based on the performance results of the 512 proxy models to predict the optimal data mixture for training large-scale LLMs. The results of the LightGBM regression analysis and Spearman correlation of the loss prediction performance are presented in 7.

Upon training the regression model, we systematically investigate the entire spectrum of potential data mixtures by utilizing the trained model to predict the target values for each candidate mixture. This process allows us to identify the input that produces the optimal target value. Following the simulation and identification of the most effective data mixture, we then generalize this top-ranked configuration for large-scale model training, incorporating a significantly larger volume of tokens.

### A.8 Full Experimental Results

We show the full results of all tasks in Table 10, Table 11 and Table 12. In analyzing the full experiment results, it is evident that our model consistently outperforms other methods across various tasks. Overall, for the zero-shot scenario, the classifier method outperforms the influence function in terms of average performance, while domain mixing yields the poorest results. Our method achieves an impressive average accuracy of 36.5, significantly surpassing the next best classifier, QuRating's series, which scores 35.5. This underscores the robustness of our approach, particularly in challenging problem-solving domains such as ARC-C, ARC-E, and MMLU, where we exceed competing models by considerable margins.

Our model demonstrates superior performance in the three-shot scenario, achieving an impressive average accuracy of 37.7, thereby maintaining its lead. Notably, we excel in the ARC-E and ARC-C benchmarks, attaining scores of 65.8 and 31.5, respectively, which highlights our model's effective

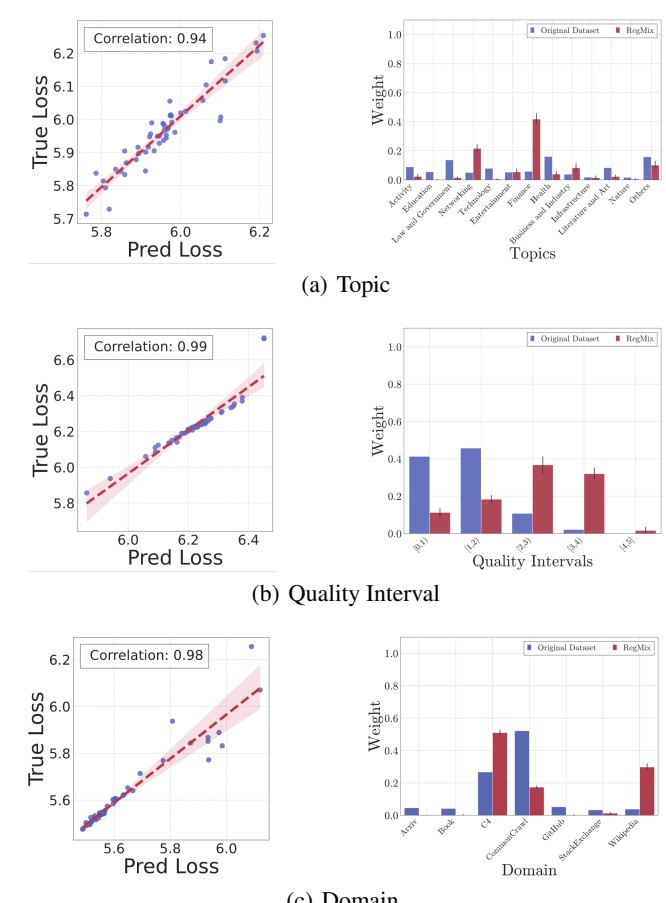

Figure 7: We present the results of the LightGBM regression analysis and Spearman correlation regarding the loss prediction performance and the weights of each candidate data-(a) Topic, (b) Quality Interval and (c) Domain after mixture across all categories.

utilization of few-shot learning. In contrast, the leading alternative methods underperform, particularly in more complex tasks such as MMLU and BoolQ.

In the five-shots evaluation, our model continues to demonstrate competitive performance, with scores reflecting a consistent trend of superiority across various domains, while other non-leading methods also maintain high levels. These results underscore our model's robust capacity to generalize across diverse question-answering tasks, affirming its advantages over conventional classifiers and highlighting its potential for practical applications in real-world scenarios.

## A.9 IMPLEMENTATION DETAILS OF OUR METHODS

To further refine the model's performance, we calculate rewards for each sampled data point by approximating the rewards using influence functions, as shown in Equation 6. Following Engstrom, we choose LAMBADA Paperno et al. (2016), SQuAD Rajpurkar (2016) and Jeopardy Tunguz (2019) as reference tasks. We followed methods provided in Engstrom, Xia et al. and Park et al. (2023) to calculate the Hessian and the gradients in the influence functions. In our implementation, we project gradients into an 8,192-dimensional space for both the validation and training datasets. To optimize the gradient computation process, we divide each data category into eight slices, thereby enabling parallel computation across eight GPUs. Each slice contains 1,250 data points. After calculating gradients for each slice, the results are concatenated in their original sequence to ensure data integrity. This slicing strategy not only accelerates the processing by utilizing GPU parallelism but also maintains consistency in gradient calculation. Additionally, for the validation datasets, we

Table 10: Table Showing Various Selection Methods and Their Scores with Changes. We report accuracy for all tasks, and **bold** the best result in each column. Abbreviations: O.B.QA = Open-bookQA W.G. = WinoGrande, C.S.QA = CommonSenseQA, Compreh. = Comprehensions

| Selection Method | Problem Solving | | | | Commonsense Reasoning | | | | Reading Compreh. | | |
|---|---|---|---|---|---|---|---|---|---|---|---|
| | ARC-E | ARC-C | MathQA | MMLU | O.B.QA | SIQA | W.G. | C.S.QA | BoolQ | RACE | Average |
| **Random sample** | | | | | | | | | | | |
| Uniform-30B | 54.3 | 23.4 | 22.3 | 23.9 | 18.6 | 39.8 | 52.8 | 19.2 | 55.4 | 30.0 | 34.0 |
| Uniform-60B | 55.2 | 24.6 | 22.5 | 23.4 | 21.0 | 39.7 | 51.9 | 19.5 | 59.8 | **33.1** | 35.1 |
| **Perplexity-based data selection** | | | | | | | | | | | |
| PPL | 49.3 | 20.1 | 22.4 | 23.6 | 16.2 | 36.0 | 48.1 | 18.8 | 61.4 | 29.3 | 32.5 |
| **Classifier-based data selection** | | | | | | | | | | | |
| QuRating-Facts | 56.1 | 23.3 | 22.4 | 24.8 | 21.6 | 39.2 | 54.1 | 19.9 | 61.5 | 31.6 | 35.5 |
| QuRating-Req | 54.9 | 24.4 | 23.2 | 25.2 | 21.4 | 38.1 | 54.5 | 20.6 | **61.6** | 31.3 | 35.5 |
| QuRating-Writing | 53.6 | 23.2 | **23.4** | 23.2 | 21.0 | 38.1 | 52.8 | 19.7 | 59.4 | 31.6 | 34.6 |
| QuRating-Edu | 57.0 | 24.4 | 22.0 | 25.0 | 20.4 | **40.3** | 53.7 | 20.2 | 60.1 | 32.2 | 35.5 |
| FineWeb-Edu | 53.8 | 23.4 | 21.8 | 23.9 | 19.8 | 39.2 | 51.7 | 20.8 | 59.7 | 32.0 | 34.6 |
| DSIR-Book | 45.4 | 20.8 | 22.0 | 23.0 | 18.8 | 39.9 | **54.6** | 19.7 | 58.3 | 30.8 | 33.3 |
| DSIR-Wiki | 50.6 | 21.1 | 21.6 | 23.0 | 19.2 | 36.6 | 53.0 | 19.8 | 60.5 | 29.2 | 33.5 |
| **Domain mixing methods** | | | | | | | | | | | |
| DOGE | 49.4 | 21.8 | 22.5 | 23.0 | 18.0 | 38.0 | 52.7 | 19.9 | 60.0 | 30.0 | 33.5 |
| DoReMi | 50.1 | 20.2 | 22.5 | 23.7 | 17.8 | 38.7 | 52.8 | 19.7 | 58.6 | 30.8 | 33.5 |
| DMLaw | 49.6 | 21.9 | 23.2 | 23.6 | 17.8 | 38.6 | 51.8 | 20.1 | 60.4 | 29.0 | 33.6 |
| RegMix | 50.0 | 22.3 | 22.1 | 22.9 | 18.8 | 38.0 | 52.8 | 19.9 | 58.9 | 31.2 | 33.7 |
| **Influence functions** | | | | | | | | | | | |
| MATES | 50.0 | 21.4 | 22.7 | 25.3 | 19.0 | 39.8 | 53.6 | **21.3** | 59.9 | 32.1 | 34.5 |
| **Multi-Agent Collaboration (Ours)** | **61.1** | **28.2** | 22.6 | **26.0** | **24.4** | 38.2 | 54.2 | 19.5 | 61.0 | 29.8 | **36.5** |

Table 11: Table showing various selection methods and their three-shots performance. We report accuracy for all tasks, and **bold** the best result in each column. Abbreviations: O.B.QA = Open-bookQA W.G. = WinoGrande, C.S.QA = CommonSenseQA, Compreh. = Comprehensions

| Selection Method | Problem Solving | | | | Commonsense Reasoning | | | | Reading Compreh. | | |
|---|---|---|---|---|---|---|---|---|---|---|---|
| | ARC-E | ARC-C | MathQA | MMLU | O.B.QA | SIQA | W.G. | C.S.QA | BoolQ | RACE | Average |
| **Random sample** | | | | | | | | | | | |
| Uniform-30B | 54.6 | 23.0 | 22.1 | 24.9 | 18.8 | 40.3 | 52.9 | **21.5** | 53.0 | 29.8 | 34.1 |
| Uniform-60B | 58.8 | 25.5 | 23.0 | **27.2** | 20.0 | **41.8** | 53.6 | 19.6 | 56.9 | **32.7** | 35.9 |
| **Perplexity-based data selection** | | | | | | | | | | | |
| PPL | 50.6 | 21.3 | 22.7 | 25.2 | 15.6 | 37.7 | 48.9 | 20.1 | 61.5 | 22.3 | 32.6 |
| **Classifier-based data selection** | | | | | | | | | | | |
| QuRating-Facts | 59.5 | 25.7 | 22.6 | 25.9 | 19.8 | 40.2 | **54.6** | 19.2 | 60.8 | 24.8 | 35.3 |
| QuRating-Req | 59.3 | 25.9 | 22.7 | 26.1 | 19.6 | 39.7 | 53.7 | 20.5 | 58.5 | 22.7 | 34.9 |
| QuRating-Writing | 56.9 | 25.7 | **23.1** | 26.0 | 20.4 | 41.1 | 53.6 | 20.2 | 51.4 | 22.6 | 34.1 |
| QuRating-Edu | 60.8 | 26.5 | 22.5 | 26.7 | 20.2 | 41.4 | 54.6 | 20.6 | 55.5 | 22.7 | 35.1 |
| FineWeb-Edu | 56.2 | 25.7 | 22.3 | 26.2 | 20.6 | 40.1 | 50.5 | 19.7 | 56.6 | 31.4 | 34.9 |
| DSIR-Book | 48.7 | 21.0 | 22.6 | 25.6 | 18.6 | 42.5 | 53.7 | 19.5 | 57.9 | 22.9 | 33.3 |
| DSIR-Wiki | 53.2 | 22.4 | 22.6 | 25.3 | 17.6 | 37.1 | 52.7 | 21.4 | **61.6** | 24.2 | 33.8 |
| **Domain mixing methods** | | | | | | | | | | | |
| DOGE | 52.4 | 21.9 | 22.4 | 27.0 | 17.4 | 39.9 | 52.0 | 18.2 | 57.8 | 29.8 | 33.9 |
| DoReMi | 53.2 | 21.4 | 22.2 | 24.7 | 18.2 | 38.4 | 50.9 | 20.6 | 59.7 | 31.1 | 34.0 |
| DMLaw | 51.5 | 21.4 | 22.4 | 25.2 | 18.2 | 39.0 | 50.7 | 19.4 | 52.6 | 29.8 | 33.0 |
| RegMix | 53.1 | 22.1 | 22.2 | 25.4 | 19.0 | 39.1 | 53.5 | 18.4 | 60.7 | 30.0 | 34.4 |
| **Influence functions** | | | | | | | | | | | |
| MATES | 52.6 | 21.8 | 22.6 | 26.7 | 20.4 | 40.9 | 53.7 | 19.7 | 57.6 | 31.8 | 34.8 |
| **Multi-Agent Collaboration (Ours)** | **65.8** | **31.5** | 23.0 | 26.6 | **24.6** | 39.9 | 54.1 | 20.1 | 60.4 | 30.5 | **37.7** |

uniformly sample 500 data points to ensure a balanced evaluation procedure. All prompts across the datasets are carefully aligned to maintain task coherence, a crucial factor in multi-task learning scenarios. Furthermore, we implement a sliding window of 1,024 tokens with a 256-token overlap to ensure consistent tokenization across the entire dataset. This sliding window technique efficiently extracts a maximum of 1,024 tokens from each data point, ensuring uniform encoding across different datasets and tasks, thus improving the overall consistency and reliability of the data processing pipeline.

Table 12: Table showing various selection methods and their five-shots performance. We report accuracy for all tasks, and **bold** the best result in each column. Abbreviations: O.B.QA = OpenbookQA W.G. = WinoGrande, C.S.QA = CommonSenseQA, Compreh. = Comprehensions

| Selection Method | Problem Solving | | | | Commonsense Reasoning | | | | Reading Compreh. | | |
|---|---|---|---|---|---|---|---|---|---|---|---|
| | ARC-E | ARC-C | MathQA | MMLU | O.B.QA | SIQA | W.G. | C.S.QA | BoolQ | RACE | Average |
| **Random sample** | | | | | | | | | | | |
| Uniform-30B | 54.5 | 21.9 | 22.4 | 25.6 | 19.2 | 39.7 | 54.2 | 19.7 | 53.2 | 30.8 | 34.1 |
| Uniform-60B | 59.1 | 26.0 | 22.4 | **26.9** | 21.6 | **42.1** | 54.3 | 21.0 | 55.7 | **32.4** | 36.2 |
| **Perplexity-based data selection** | | | | | | | | | | | |
| PPL | 49.2 | 21.2 | 22.5 | 24.9 | 14.6 | 36.7 | 49.8 | 20.6 | 60.6 | 23.3 | 32.4 |
| **Classifier-based data selection** | | | | | | | | | | | |
| QuRating-Facts | 60.5 | 25.4 | **23.4** | 26.4 | 20.2 | 40.3 | 51.9 | 19.0 | 58.0 | 23.3 | 34.8 |
| QuRating-Req | 59.9 | 26.4 | 22.8 | 25.6 | 21.8 | 40.1 | 53.7 | 19.6 | 56.9 | 22.3 | 34.9 |
| QuRating-Writing | 57.3 | 25.3 | 22.6 | 25.0 | 21.4 | 41.6 | 53.5 | 19.6 | 49.5 | 22.1 | 33.8 |
| QuRating-Edu | 60.8 | 26.5 | 22.5 | 26.5 | 20.2 | 41.4 | **54.6** | **21.1** | 55.5 | 22.7 | 35.2 |
| FineWeb-Edu | 56.6 | 24.9 | 22.6 | 25.8 | 19.8 | 39.4 | 51.2 | 19.7 | 55.9 | 30.9 | 34.7 |
| DSIR-Book | 49.7 | 21.1 | 22.1 | 25.6 | 19.8 | 41.7 | 54.1 | 18.3 | 55.6 | 22.9 | 33.1 |
| DSIR-Wiki | 53.6 | 22.3 | 23.0 | 25.3 | 17.6 | 36.7 | 52.2 | 20.4 | 60.2 | 22.6 | 33.4 |
| **Domain mixing methods** | | | | | | | | | | | |
| DOGE | 53.0 | 21.8 | 22.0 | 26.3 | 17.2 | 40.1 | 51.7 | 18.8 | 58.5 | 30.1 | 33.9 |
| DoReMi | 52.7 | 22.2 | 22.4 | 25.5 | 16.2 | 39.3 | 51.9 | 20.9 | 60.0 | 31.0 | 34.2 |
| DMLaw | 52.4 | 21.4 | 23.0 | 25.7 | 17.2 | 39.2 | 50.6 | 19.2 | 51.4 | 29.9 | 33.0 |
| RegMix | 53.5 | 24.0 | 21.2 | 25.0 | 19.6 | 41.0 | 53.2 | 19.0 | **61.3** | 30.2 | 34.8 |
| **Influence functions** | | | | | | | | | | | |
| MATES | 53.6 | 21.6 | 22.6 | 26.1 | 20.4 | 41.7 | 53.1 | 20.4 | 60.1 | 32.0 | 35.2 |
| **Multi-Agent Collaboration (Ours)** | **64.9** | **31.1** | 22.4 | 26.3 | **23.6** | 39.0 | 53.1 | 20.4 | 60.4 | 30.7 | **37.2** |

## A.10 DETAILS OF INFLUENCE CHANGES DURING DIFFERENT PRETRAINING STAGES

We present the details of influence change during the pretraining process for domain (Figure 8), quality intervals (Figure 9) and topic (Figure 10).

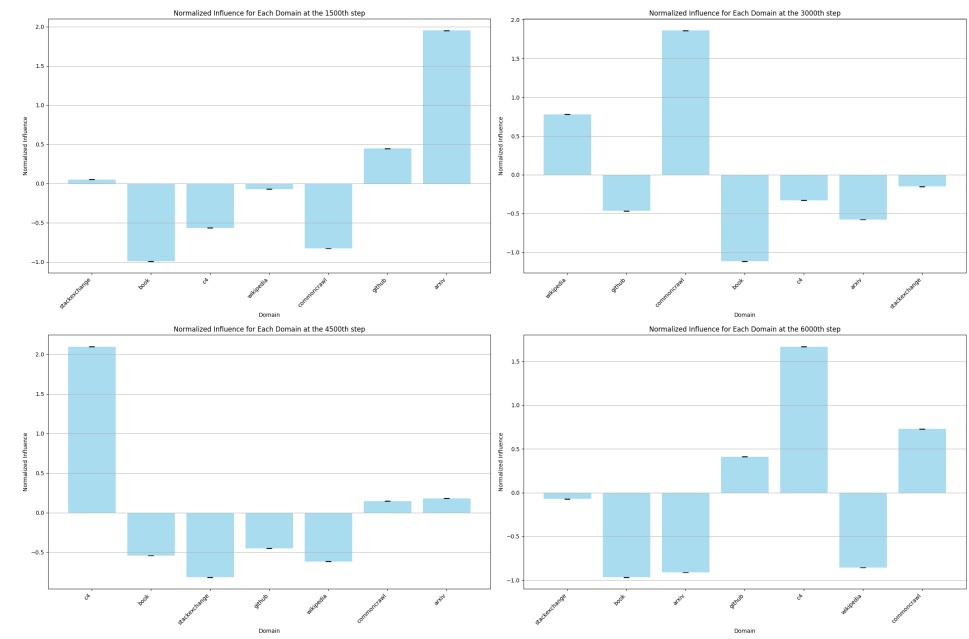

Figure 8: We present the normalized influence for each domain across various training steps.

## B DATA DISTRIBUTION ANALYSIS OF THE SLIMPAJAMA DATASET

We finally present the data distribution analysis of the SlimPajama dataset from three dimensions: topic, domain and quality intervals, as Figure 11 to Figure 13 shows.

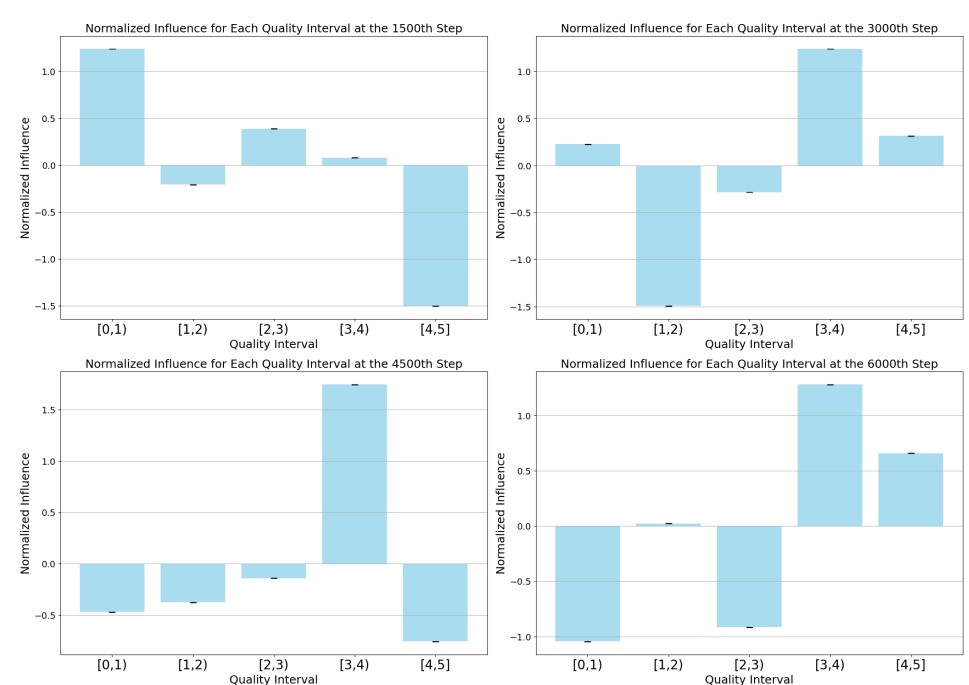

Figure 9: We present the normalized influence for each quality interval across various training steps.

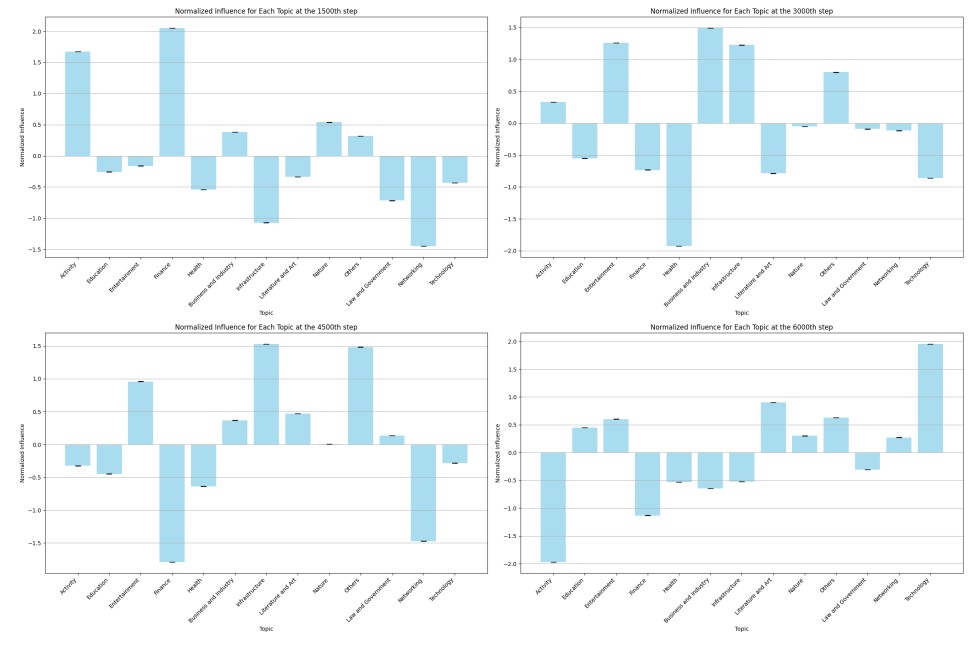

Figure 10: We present the normalized influence for each topic across various training steps.

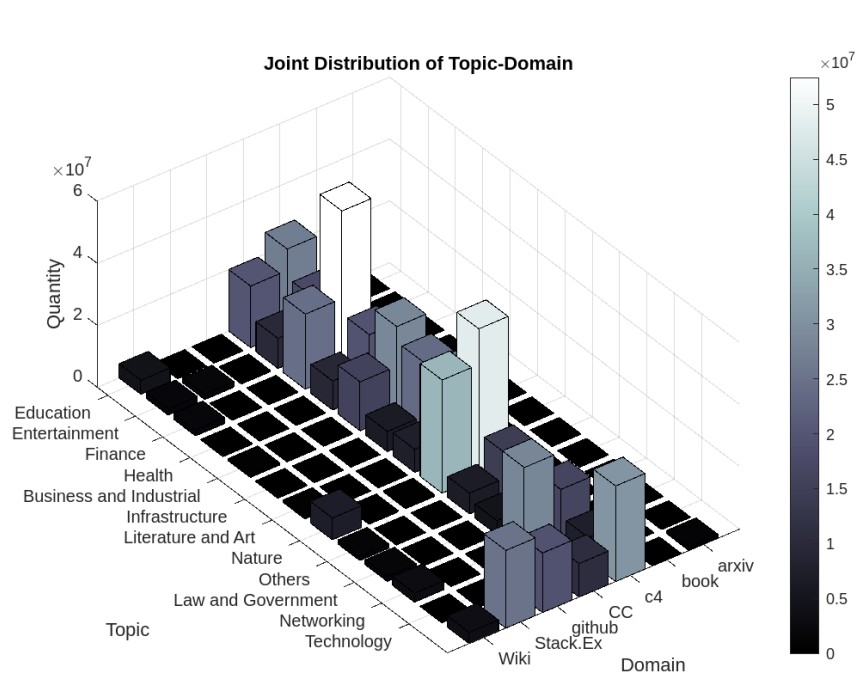

Figure 11: The illustration of the joint distribution of topics and domains.

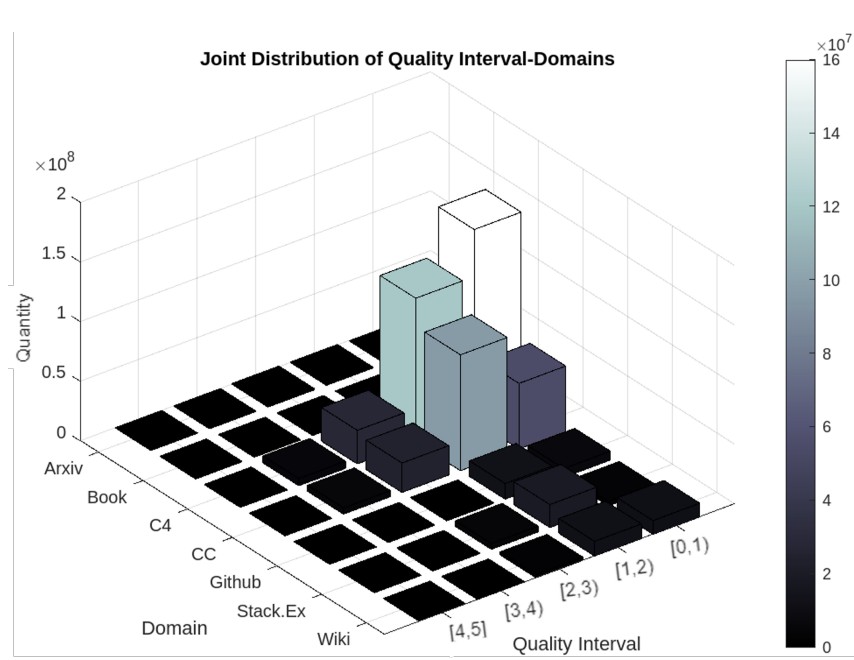

Figure 12: The illustration of the joint distribution of quality intervals and domains.

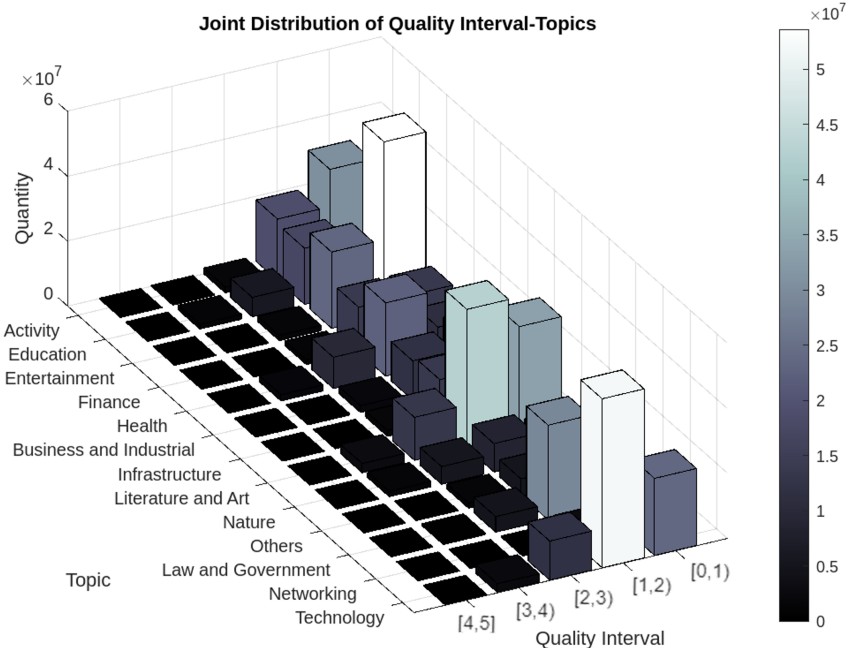

Figure 13: The illustration of the joint distribution of quality intervals and topics.

