# OpenReview forum: "Multi-Agent Collaborative Data Selection for Efficient Language Model Pretraining"
_ICLR.cc/2025/Conference — Submitted to ICLR 2025_

### Official Review · Reviewer_6r7V · 2024-10-26

**Soundness:** 2
**Presentation:** 3
**Contribution:** 2
**Rating:** 3
**Confidence:** 4

**Summary:**

The authors argue that training data selection is an important component in LLM training. The various techniques that had been proposed might be conflicting in their recommendations. The authors are proposing a technique in which the different data selection algorithms are considered independent agents, with an "agent console" integrating the recommendations. The approach enable the dynamic adjustment of contributions of the agents during the training process of the LLM. The SlimPajama dataset is used as the working example.

**Strengths:**

* A case study validates the initial claim of the paper, that the quality, diversity and influence scores of data are not strongly correlated with each other.
* Relatively extensive experiments were conducted with the training of a 1.3B LLAMA LLM.

**Weaknesses:**

* The fact that the diversity of topics, quality of material, and influence on the trained LLMs are different metrics is not a surprising observation - these are obviously very different things. A strong correlation between them would be the surprise.
* The term "agent" has a relatively clear definition in the AI literature, as an autonomous entity, that takes actions in an environment, in the pursuit of a goal. The fact that the authors have to introduce a definition for the term "agent" and "agent console" and define them in terms of "data selection method" makes the paper difficult to follow. It doesn't seem that these "agents" are taking any actions, or have any autonomy.
* In practice, the proposed approach appears to be a way to make a decision about what training data to include based on weighting three pre-existing metrics about the data sources. This decision process could have been easily written without introducing agent language. In fact, practitioners very likely are already making such judgements.

Considerations after reading the answers: despite the authors providing very long rebuttals, the answers contain nothing concrete that would change the evaluation.

**Questions:**

* How does the definition of the agent used in this paper relates to the term agent used in other fields of AI? Such as multi-agent RL, or the type of work usually published in AAMAS?
* The term "console" is usually applied to a component that is used by a human operator. Is the term agent console in the paper related to it?

NOTE about answers:
- This reviewer is obviously familiar with the Russel-Norvig book, that overall takes an agent view to AI. Nevertheless, what the authors propose here are not agents, as pointed out by other reviewers as well.
- The answers contain references to the consoles in Salesforce etc. But if the authors would have checked what those products are, they are intended for the human operator.

---

> ### Author Response · Authors · 2024-11-22
> **Authors Response 1/2**
>
> Thank you for the efforts in reviewing our paper!
>
> > W1: The fact that the diversity of topics, quality of material, and influence on the trained LLMs are different metrics is not a surprising observation - these are obviously very different things. A strong correlation between them would be the surprise.
> >
>
> Thank you for this insightful point! We completely agree that diversity of topics and domains, quality of material, and influence on trained LLMs are distinct metrics. Our surprise does not lie in recognizing these as fundamentally different aspects. Instead, it stems from the fact that *despite the inherent conflicts and differences among these criteria, studies have shown that using any one of them as a standalone data selection criterion can still lead to good data selection trajectories* *for convergence* [1, 2, 3]. This underscores the importance of understanding how to effectively integrate these differences, which we believe is the truly surprising finding.
>
> **In the updated draft, we have revised the introduction to address your concerns and clarify our motivation more effectively.**
>
> [1] Xie S M, Pham H, Dong X, et al. Doremi: Optimizing data mixtures speeds up language model pretraining. NeurIPS, 2023.
>
> [2] Wettig A, Gupta A, Malik S, et al. QuRating: Selecting High-Quality Data for Training Language Models. ICML, 2024.
>
> [3] Yu Z, Das S, Xiong C. MATES: Model-Aware Data Selection for Efficient Pretraining with Data Influence Models. NeurIPS, 2024.
>
> > W2: The term "agent" has a relatively clear definition in the AI literature, as an autonomous entity, that takes actions in an environment, in the pursuit of a goal. The fact that the authors have to introduce a definition for the term "agent" and "agent console" and define them in terms of "data selection method" makes the paper difficult to follow. It doesn't seem that these "agents" are taking any actions, or have any autonomy.
> >
>
> Thank you for highlighting this point. While we recognize that the term "agent" is broadly defined in AI literature, its interpretation often varies across application domains. For example, our agent is different from LLM-based agent in other research in the LLM community that focuses on handling complex language model reasoning tasks, as seen in [1]. In our scenario, the agent generally refers to an entity that perceives some status and map the observed status into actions according to the classic definition of intelligent agent in [2].
>
> Concretely, the observed status of the environment refers to the current state of the pretraining model. During each stage of data selection, agents take actions sample data and adjust their internal weights based on the observed status, prioritizing the good data according to the updated weights. The agent console dynamically updates the contribution of each agent based on the model's observed state, ensuring a balance in the prioritization of data selected by different agents. This coordinated process leads to data selection for subsequent training. Throughout this workflow, the agent observes the current state of model training and maps this status into its individual data selection decisions. These decisions are then integrated into the final selection process, effectively contributing to a well-structured formulation of the data selection mechanism.
>
> **We have slightly revised the language in our paper to make the formulation clearer, with the changes highlighted in blue. Additionally, we have added Appendix A.3, which includes detailed mathematical formulations.**
>
> [1] Hong S, Zhuge M, Chen J, et al. MetaGPT: Meta Programming for A Multi-Agent Collaborative Framework. ICLR, 2024.
>
> [2] Russell S J, Norvig P. Artificial intelligence: a modern approach. Pearson, 2016.

---

> > ### Comment · Reviewer_6r7V · 2024-11-28
> >
> > The authors answers do not change my original evaluation.

---

> ### Author Response · Authors · 2024-11-22
> **Authors Response 2/2**
>
> > W3: In practice, the proposed approach appears to be a way to make a decision about what training data to include based on weighting three pre-existing metrics about the data sources. This decision process could have been easily written without introducing agent language. In fact, practitioners very likely are already making such judgements.
> >
>
> Thank you for highlighting this important point! Comparing with the fixed weighting strategies on three pre-existing metrics about the data sources to make data selection decision, as we explained in the response of W2, the power of our framework is that our agent and agent console dynamically capture the status of current environment (which is the current model states) to updates its data selection decisions. The agent formulation provides a good formulation about this.
>
> Empirically, as shown in the Ablation Study in Table 2, we compared our approach which utilizes dynamic agent collaboration weights, to a method employing a fixed averaging of those criterion. The detailed results for each training stage are presented in the following table. Our dynamic adjustment method consistently outperforms the fixed-weight approach, delivering an overall performance improvement of 7.0% compare with the fixed heuristic-based solution to determine the weights with different preexisting metrics.
>
> | Training steps | 1500 | 3000 | 4500 | 6000 | 7500 |
> | --- | --- | --- | --- | --- | --- |
> | with dynamic adjustment | 31.1 | **33.3** | **35.9** | **36.7** | **37.7** |
> | without dynamic adjustment | 31.1 | 32.9 | 33.6 | 34.4 | 35.1 |
>
> > Q1: How does the definition of the agent used in this paper relates to the term agent used in other fields of AI? Such as multi-agent RL, or the type of work usually published in AAMAS?
> >
>
> Thank you for the questions! As we explained in W2, our agent is inspired by the classic definition of the intelligent agent outlined in [1], where an agent is broadly defined as an entity that perceives a given state and maps its observations to corresponding actions.
>
> We appreciate the reviewer pointing out lots of  fascinating works on multi-agent reinforcement learning. We tend to believe these methods are tangential to the focus of our study. While we use a similar term, it refers to a different line of research. **Additionally, we have added Appendix A.3, which includes detailed comparisons between our methods and traditional multi-agent RL formulations.**
>
> [1] Russell S J, Norvig P. Artificial intelligence: a modern approach. Pearson, 2016.
>
> > Q2: The term "console" is usually applied to a component that is used by a human operator. Is the term agent console in the paper related to it?
> >
>
> Thank you for the questions. In this paper, the term "agent console" refers to a role that integrates outputs from multiple agents to make final decision. From our initial perspective, this term can be used to refer the common use of "console" as a tool that aggregates inputs for efficient decision-making. For example, platforms like Salesforce's Service Cloud[1] provide an "Agent Console" for managing customer interactions, while Google's Dialogflow CX[2] offers an "Agent Builder Console" to design and oversee conversational agents.
>
> We realized the potential risk of using this term, and we would be happy to make the modification if the reviewer would like to suggest a more precise name.
>
> [1] https://www.salesforce.com/service/cloud/
>
> [2] https://cloud.google.com/dialogflow/cx/docs/concept/builder-console?hl=

---

> ### Author Response · Authors · 2024-11-25
>
> Dear Reviewer 6r7V,
>
> We sincerely thank you once again for the invaluable time and effort you have dedicated to reviewing our work.
>
> To address your feedback on our motivations, we have revised the draft for clarity. For concerns about our multi-agent formulation, we included detailed comparisons to multi-agent RL in Appendix A.3 and refined the definition for better clarity. We hope our response, along with the additional results and revised draft, address all your questions and concerns.
>
> As the discussion period nears its conclusion, we would greatly appreciate it if you could review our reply at your convenience. If there are any remaining concerns we haven't addressed yet for a better score, we are committed to addressing them to the best of our ability.
>
> We are deeply grateful for your patience and insightful contributions, and we sincerely look forward to further discussion!

---

> ### Author Response · Authors · 2024-11-29
>
> Dear Reviewer 6r7V,
>
> Thank you for your comments and feedback, and happy Thanksgiving!
>
> We greatly appreciate the time and effort you took to evaluate our work. **In response to your concerns, which were primarily focused on issues in the writing of our draft, we have carefully revised our manuscript to address each point you raised.** We believe these updates have made the writing clearer and more concise.
>
> While we recognize that there were aspects of our initial draft that could be improved in terms of writing, we also believe, **as highlighted by other reviewers, that our methods present significant contributions and potential for impact.** If our current revisions have not fully addressed your concerns, we kindly ask if you could provide more specific or concrete feedback. This would help us better understand and address the issues that led to the current evaluation, particularly the score of 3.
>
> We believe that the OpenReview platform was designed to foster meaningful dialogue and constructive exchanges between reviewers and authors, creating a community that provides valuable suggestions for improving submissions. With that in mind, **we sincerely hope that you can offer more detailed feedback that would be helpful in further refining our work.**
>
> We are committed to improving our work and will do our utmost to resolve any outstanding concerns you might have.
>
> Thank you once again for your feedback and for considering our request.
>
> Best regards,
>
> ICLR Authors

---

> ### Author Response · Authors · 2024-12-02
>
> Dear Reviewer 6r7V,
>
> Thank you for your thoughtful efforts in reviewing our work! As the discussion period approaches its conclusion, we kindly hope you can provide more specific concerns regarding our work to help us address them comprehensively.
>
> To address the points you raised in the original review:
>
> + *Case study in our draft*: We have revised the introduction and abstract to clarify the goal of our case study. As noted by other reviewers (Reviewer dSQq and WfeZ), **the conflicts among different SOTA data selection methods remain underexplored, presenting a strong motivation for developing our approach.**
>
> + *Agent terminology usage*: In response to your concerns, we added further explanation regarding the origin and definition of the term "agent" in our revised draft. We also included a mathematical comparison between our agent and the "multi-agent RL" paradigm you mentioned, provided in Appendix A.3. Importantly, **our use of "agent" adheres to established concepts from classical AI literature, avoiding unconventional terminology**. Moreover, **the core methodologies and experimental results—the main contributions of our work—remain unaffected by the terminology employed.**
>
> + *Comparison of dynamic adjustment vs. predefined fixed weights*: We added detailed experimental results demonstrating the superior performance of our dynamic adjustments through agent updates and collaboration compared to fixed weights for different data selection metrics. **Our methods consistently outperform fixed-weight approaches at each step**, underscoring the necessity of dynamically adjusting agent weights and collaborative interactions throughout the training process.
>
> We sincerely appreciate your review and hope our responses, along with the additional experiments, address your concerns. Your insights are invaluable, and we would greatly value further suggestions on how we might refine our work to **merit a higher evaluation**.
>
> Looking forward to your feedback!
>
> Best regards,
>
> ICLR Author

---

> ### Author Response · Authors · 2024-12-03
>
> Dear Reviewer 6r7V,
>
> Thank you for your valuable efforts in reviewing our work! We have thoroughly addressed all your concerns and questions, resolving the issues you raised regarding our paper. With the discussion period concluding in **9 hours**, we kindly request you to give out some concrete concerns or suggestions to our draft and response.
>
> We look forward to your feedback!
>
> Best regards,
>
> ICLR Author

---

> ### Author Response · Authors · 2024-12-03
>
> Dear Reviewer 6r7V,
>
> Thank you for your thoughtful and valuable feedback on our work! We have conducted additional experiments and made revisions to our paper to address the questions and concerns you raised. We hope these updates effectively address your points.
>
> If you have any further concrete concerns, we will carefully address them in the final version of our paper.
>
> We kindly hope you take a moment to review our responses and give some concrete suggestions or concerns.
>
> Best regards,
>
> ICLR Authors

---

> ### Author Response · Authors · 2024-12-03
> **Response to reviewer's update review after the author-reviewer discussion period**
>
> Thank you for your update of the review *after the author-reviewer discussion period*. For your updated questions we have the following responses:
>
> > Q3: This reviewer is obviously familiar with the Russel-Norvig book, that overall takes an agent view to AI. Nevertheless, what the authors propose here are not agents, as pointed out by other reviewers as well.
> >
>
> You acknowledge our familiarity with **Artificial Intelligence: A Modern Approach** [1], so it should be evident that our definition is derived from this book, and all aspects of our agent design are inspired by its principles. While we recognize that our original submission lacked sufficient detail regarding our agent's operation and its distinctions from those in "multi-agent RL," we have carefully revised our paper. Specifically, we have added precise mathematical definitions for all the agent formulation, as well as a comparison to "multi-agent RL" in Appendix A.3 according to your previous concerns.
>
> As stated in Chapter 2 of the book [1]:
>
> *"An agent is anything that can be viewed as perceiving its environment through sensors and acting upon that environment through actuators."*
>
> Our agent was designed based on this definition and the accompanying explanations in the text. It closely resembles the model-based reflex agent. We provide a concrete explanation of our agent using the pseudocode for the **model-based reflex agent from Section 2.4 (Page 53 of [1]):**
>
>
> ```jsx
> function MODEL-BASED-REFLEX-AGENT(percept) returns an action
> persistent:
> STATE, the agent’s current conception of the world state // The internal state is the current memory and weight of each agent
> TRANSITION MODEL, a description of how the next state depends on the current state and action // This is how the agent updates its memory and weight, which is defined by Equation 5 in our work.
> SENSOR MODEL, a description of how the current world state is reflected in the agent’s percepts // In our work, this perception process is structured into several steps: (1) sampling data, (2) interacting with the current model to score the sampled data, and (3) receiving feedback from the environment. These steps define how the agent perceives the current state of the world. After perceiving the current state, the data selection agent updates its internal memory and weights accordingly.
> RULES, a set of condition–action rules // Which is the predefined data selection metric for each agents
> ACTION, the most recent action, initially none  // The data selection agent prioritize the good data according to its internal weights and memories
> state ← UPDATE-STATE(state, action, percept, transition model, sensor model)
> rule ← RULE-MATCH(state, rules)
> action ← rule.ACTION
> return action
> ```
>
> The algorithm above clearly demonstrates how our agent adheres to the agent definition provided in Chapter 2 of [1]. *We have included a similar explanation in the revised version of Appendix A.3 of our paper.*
>
> **If you believe that our agent does not conform to the definition in the book, we sincerely request that you provide specific evidence of which aspects do not align instead of claiming our agents are not agent without any concrete reasons.**
>
>
> [1] Russell S J, Norvig P. Artificial intelligence: a modern approach[M]. Pearson, 2016.
>
>
> > Q4: The answers contain references to the consoles in Salesforce etc. But if the authors would have checked what those products are, they are intended for the human operator.
> >
>
> In our previous response, we clearly demonstrated that our 'agent console' is inspired by tools designed to aggregate inputs for efficient decision-making. We kindly requested your suggestions for potential modifications and still welcome any alternative suggestions for this term.

---

> ### Author Response · Authors · 2024-12-03
>
> > W4: Considerations after reading the answers: despite the authors providing very long rebuttals, the answers contain nothing concrete that would change the evaluation.
> >
> To address the points raised in your original review, we provide the following **concrete responses, supported by additional experimental results and a revised paper:**
>
> + Case study in our draft: We have **revised the introduction and abstract** to clarify the goal of our case study. As noted by other reviewers (Reviewer dSQq and WfeZ), the conflicts among different SOTA data selection methods remain underexplored, presenting a strong motivation for developing our approach.
>
> + Agent terminology usage: In response to your concerns, we **added further explanation regarding the origin and definition of the term "agent" in our revised draft**. We also **included a mathematical comparison between our agent and the "multi-agent RL" paradigm you mentioned, provided in Appendix A.3**. Importantly, our use of "agent" adheres to established concepts from classical AI literature, avoiding unconventional terminology. Moreover, the core methodologies and experimental results—the main contributions of our work—remain unaffected by the terminology employed.
>
> + Comparison of dynamic adjustment vs. predefined fixed weights: We **added detailed experimental results** demonstrating the superior performance of our dynamic adjustments through agent updates and collaboration compared to fixed weights for different data selection metrics. Our methods consistently outperform fixed-weight approaches at each step, underscoring the necessity of dynamically adjusting agent weights and collaborative interactions throughout the training process.
>
> We have included additional results and carefully revised our paper in response to your previous evaluation. We are puzzled by the statement that "the answers contain nothing concrete that would change the evaluation." **We sincerely hope you can provide specific reasons and evidence to support this assessment, rather than repeating the earlier remark that "the authors’ answers do not change my original evaluation."**

---

### Official Review · Reviewer_dSQq · 2024-11-02

**Soundness:** 3
**Presentation:** 3
**Contribution:** 3
**Rating:** 6
**Confidence:** 3

**Summary:**

This paper presents a multi-agent approach to strategically select data for pretraining LLMs. The paper motivates the need for a multi-agent design by sharing a case study on the SlimPajama pretraining dataset. They illustrate that the most common data set considerations (and their corresponding metrics), including data quality, topic diversity, data impact, and data domain are not straightforward to jointly optimize. Therefore, they propose a multi-agent system, where each data selection method/metric is represented as an agent. Through this multi-agent approach, these methods can be mixed via multi-agent collaboration, forming a highly adaptive approach in data selection. They show that their multi-agent approach is effective: the data curated by their method leads to faster convergence in training and improved benchmark performance.

**Strengths:**

-Mixing data quality and data selection techniques is a challenging problem: they show in their case study that typical data curation techniques can conflict and naively combining them is not sufficient.

-Unlike many off-the-shelf multi-agent systems, they are proposing optimization of each agent’s weights (stored in memory) based on reward signals from the model undergoing pretraining.

-Results show that their multi-agent data selection produces the best performance. Ablations show the three agents in collaboration outperform all other permutations of the agents (with and without collaboration) and strongly outperform the setting with no agents at all.

Originality:
I am not aware of another multi-agent approach for data selection in pretraining - so this appears to be a novel application of multi-agent systems.

Quality:
All key components covered - clear literature review, motivation, experiment design, results. It would have been better if they made their contribution differentiations clear in the literature review - for ex, confirming if they are indeed the first multi-agent approach for data selection. And if not, how are they different.

Clarity:
The paper was generally well written and easy to read.

Significance:
Pretraining is the most critical and expensive operation for LLMs. To make the best use of your pretraining, optimizing the data is key.

**Weaknesses:**

-Figures can be improved, see my comments below.

-Experiments only on 1B model. It would be interesting to see the impacts across more model sizes (smaller and larger) and model architectures to see which range of models really benefit from this.

-This shows a lot of potential already, but the point would be very strongly proven if they could show comparison to other 1B model performances (DeepSeek, TinyLlama, etc.), showing that their approach yields superior models in general.

**Questions:**

-Figure 2 could be more clear to show that each agent has its own memory. Consider focusing only on the Domain Agent’s flow of work to make it easier to follow.

-In Table 1, it would be best to bold the top results so it’s easier to see that your approach is indeed among the best.

-Should this really be considered a multi-agent system or is this really a multi-step process? I don’t see any use of reasoning or decision making here. It seems at each step, each agent is systemically called/updated and each data point is labeled with a combination of scores from the “agents”. What is “agentic” about this? The approach is still valuable, just questioning whether it falls under “agents”.

-Does your approach add significant additional latency to the pretraining stage?

-Is your approach particularly valuable for small models?

---

> ### Author Response · Authors · 2024-11-22
> **Authors Response 1/2**
>
> Thank you for the efforts in reviewing our paper!
>
> > W1&Q2: Figure 2 could be more clear to show that each agent has its own memory. Consider focusing only on the Domain Agent’s flow of work to make it easier to follow.
> >
>
> Thank you for the suggestion! We have revised **Figure 2** in the updated draft to clearly show that each agent has its own memory and focus on the Domain Agent’s workflow to improve clarity.
>
> > W2: Experiments only on 1B model. It would be interesting to see the impacts across more model sizes (smaller and larger) and model architectures to see which range of models really benefit from this.
> >
>
> > Q5:Is your approach particularly valuable for small models?
> >
>
> **Generalize to other model sizes and architectures.** Our main experiments and ablation studies demonstrate that, compared to random selection, our method improves overall accuracy by 12.5% on a 373M LLaMA2 model and 10.5% on a 1.3B LLaMA2 model. We present results on an additional experiment training a 3.6B LLaMA3.2 model from scratch on 36B tokens below, where our method achieves a 13.7% performance gain over random selection. This consistent trend across three different model sizes highlights the generalization capability of our method to various model scales. Furthermore, experiments on different LLaMA model architectures confirm that our method is generally applicable across different versions of the LLaMA architecture. The consistent performance gains suggest that our method is not only effective for smaller models but also holds strong potential for training much larger models, including those with 10B+ parameters. We plan to explore its applicability to larger models and alternative architectures in future work.
>
> **In our updated draft, we have added an analysis of the generalization of our methods in Appendix A.6.**
>
> |  | ARC-C | ARC-E | MathQA | MMLU | O.B.QA | SIQA | W.G. | C.S.QA | BoolQ | RACE | Average |
> | --- | --- | --- | --- | --- | --- | --- | --- | --- | --- | --- | --- |
> | 3.6B (Random) | 17.7 | 34.8 | 21.3 | 23.0 | 12.0 | 32.9 | 50.2 | 19.6 | 37.8 | 20.9 | 27.0 |
> | 3.6B (Ours) | **21.3** | **42.9** | **21.9** | **24.0** | **15.8** | **33.9** | **51.0** | **20.4** | **54.8** | **21.2** | **30.7** |
>
> > W3: This shows a lot of potential already, but the point would be very strongly proven if they could show comparison to other 1B model performances (DeepSeek, TinyLlama, etc.), showing that their approach yields superior models in general.
> >
>
> We conduct additional experiments to compare our approach with two open-source models around 1B parameters: DeepSeek[1] and TinyLlama[2]. We evaluate the three-shot performance of these models alongside our model trained on 30B randomly selected tokens and another version trained on 30B tokens curated using our data selection method. While the model trained with random token selection underperforms compared to the two open-source models, the model trained on our curated dataset achieves comparable results across all tasks and delivers the best average performance overall. Notably, *our model is trained on significantly fewer tokens than the other two: DeepSeek-coder-1.3b-base was trained on 130B natural language tokens, and TinyLlama1.1B on 105B tokens, whereas our model was trained on just 30B selected tokens.* We believe this result strongly demonstrates the effectiveness of our data selection method.
>
> |  | ARC-C | ARC-E | MathQA | MMLU | O.B.QA | SIQA | W.G. | C.S.QA | BoolQ | RACE | Average |
> | --- | --- | --- | --- | --- | --- | --- | --- | --- | --- | --- | --- |
> | Deepseek-coder-1.3B-base | 21.7 | 49.1 | **26.1** | 25.4 | 15.8 | 37.7 | 53.7 | 19.7 | **64.2** | 30.3 | 34.4 |
> | TinyLlama1.1B-105B tokens | 22.9 | 55.7 | 23.3 | **27.3** | 19.6 | **40.8** | **54.2** | 18.9 | 55 | 30.4 | 34.8 |
> | Random 1.3B-30B tokens | 23.0 | 54.6 | 22.1 | 24.9 | 18.8 | 40.3 | 52.9 | **21.5** | 53.0 | 29.8 | 34.1 |
> | Ours 1.3B-30B tokens | **31.5** | **65.8** | 23 | 26.6 | **24.6** | 39.9 | 54.1 | **20.1** | 60.4 | **30.5** | **37.7** |
>
> [1] https://huggingface.co/deepseek-ai/deepseek-coder-1.3b-base
>
> [2] https://huggingface.co/TinyLlama/TinyLlama-1.1B-intermediate-step-240k-503b
>
> > Q2: In Table 1, it would be best to bold the top results so it’s easier to see that your approach is indeed among the best.
> >
>
> Thanks for pointing this out! In the revised version of the paper, we bold the best-performing results across relevant metrics in **Table 1** to enhance readability and highlight the competitiveness of our method.

---

> ### Author Response · Authors · 2024-11-22
> **Authors Response 2/2**
>
> > Q3: Should this really be considered a multi-agent system or is this really a multi-step process? I don’t see any use of reasoning or decision making here. It seems at each step, each agent is systemically called/updated and each data point is labeled with a combination of scores from the “agents”. What is “agentic” about this? The approach is still valuable, just questioning whether it falls under “agents”.
> >
>
> Thank you for your insightful question! We acknowledge the essential difference between our work and research in the LLM community that focuses on leveraging LLM-based agents to handle complex language model reasoning tasks, as seen in [1]. In our scenario, the agent generally refers to an entity that perceives some status and map the observed status into actions according to the definition of classic intelligent agent in [2].
>
> Specifically, as stated in Chapter 2 of the book [2]: "An agent is anything that can be viewed as perceiving its environment through sensors and acting upon that environment through actuators."
> Our agent was developed in accordance with this definition and the accompanying explanations in the text. It closely resembles the model-based reflex agent outlined on Page 53 in Section 2.4 of [2], with each component of the agent directly mapped to the definition of a model-based reflex agent. Additionally, we have included a detailed functional comparison in the *Global Response.*
>
> Concretely, the observed status refers to the current state of the pretraining model. During each stage of data selection, agents sample data and adjust their internal weights based on the observed status, prioritizing the good data according to the updated weights. The agent console dynamically updates the contribution of each agent based on the model's observed state, ensuring a balance in the prioritization of data selected by different agents. This coordinated process leads to data selection for subsequent training. Throughout this workflow, the agent observes the current state of model training and maps this status into its individual data selection decisions. These decisions are then integrated into the final selection process, effectively contributing to a well-structured formulation of the data selection mechanism.
>
> **We have slightly revised the language in our paper to make the formulation clearer, with the changes highlighted in blue. Additionally, we have added Appendix A.3, which includes detailed mathematical formulations.**
>
> [1] Hong S, Zhuge M, Chen J, et al. MetaGPT: Meta Programming for A Multi-Agent Collaborative Framework. The Twelfth International Conference on Learning Representations.
>
> [2] Russell S J, Norvig P. Artificial intelligence: a modern approach[M]. Pearson, 2016.
>
> > Q4: Does your approach add significant additional latency to the pretraining stage?
> >
>
> Thank you for highlighting this important point. We would like to clarify that our approach does not significantly increase latency during the pretraining stage. Pretraining a 1.3B model on 30B tokens requires approximately $3.04\times 10^{20}$ FLOPs. Our data selection method incurs only an additional $1.19\times 10^{18}$ FLOPs for computing influence functions throughout the training process, which is negligible compared to the pretraining cost. In contrast to the online updating baseline MATES [1], which requires $1.98 \times 10^{20}$ FLOPs for both online influence function computation and labeling, our method eliminates the need to label the entire training dataset during training. This results in significantly reduced latency. **We have updated our drafts with this detailed analysis in Appendix A.4.**
>
> | Seletion Method | Online Labeling Cost (FLOPs) | Influence Score Computation Cost (FLOPs) | Overall FLOPs during online update |
> | --- | --- | --- | --- |
> | MATES[1] | $1.98 \times 10^{20}$ | $1.19\times 10^{18}$ | $1.99 \times 10^{20}$ |
> | Multi-agent collaboration (ours) | N.A. | $1.19\times 10^{18}$ | $1.19\times 10^{18}$ |
>
> [1] Yu Z, Das S, Xiong C. MATES: Model-Aware Data Selection for Efficient Pretraining with Data Influence Models. Advances in neural information processing systems, 2024.

---

> ### Author Response · Authors · 2024-11-25
>
> Dear Reviewer dSQq,
>
> We sincerely thank you once again for the invaluable time and effort you have dedicated to reviewing our work!
>
> To address your concerns on generalization of our methods and comparisons to open-source models, we added new experiments, and added analysis in Appendix A.6. For multi-agent formulation and latency problem, we added detailed explanations in Appendic A.3 and A.4 respectively. We also updated figures and tables per your suggestions. We hope our response with these new experiments and the revised draft address all your concerns.
>
> As the discussion period is nearing its conclusion, we would greatly appreciate it if you could find a moment to review our reply at your convenience. If there are any remaining concerns we haven't addressed yet for a better score, we are committed to addressing them to the best of our ability.
>
> We are deeply grateful for your patience and insightful contributions, and we sincerely look forward to further discussion!

---

> ### Author Response · Authors · 2024-12-02
>
> Dear Reviewer dSQq,
>
> Thank you for your thoughtful efforts in reviewing our work! As the discussion period nears its end, we kindly hope you can take a moment to review our rebuttal.
>
> To address the concerns and questions you raised:
>
> + *Suggestions on figures and tables*: We have carefully **revised Figure 2 and Table 1** in line with your suggestions and added bold text to enhance clarity **across all tables**.
>
> + *Scalability across model sizes and architectures*: We conducted **additional experiments on 3.6B and 8B LLaMA3.2 architecture models**. This extends our analysis to four model sizes (373M, 1.3B, 3.6B, and 8B parameters) and includes comparisons across various LLaMA versions (LLaMA2 and LLaMA3.2). Our results **consistently demonstrate over a 10% average performance improvement** compared to random selection across widely-used benchmarks, underscoring the scalability, generalizability, and impact of our methods.
>
> + *Comparison with other open-source models*: We added results for other open-source models, showing that our methods outperform these models, even those trained on more tokens, on average. This highlights the effectiveness of our data selection method.
>
> + *Additional latency*: A comparison of the additional latency introduced by our methods versus other SOTA approaches is provided in Appendix A.4. Our methods incur **minimal additional latency** compared to other online methods.
>
> + *Multi-agent paradigm*: We compared our multi-agent paradigm to multi-step processes (or multi-step optimization processes) **in Appendix A.3**. We have also revised the paper to more clearly articulate the "agentic" nature of our framework. We show the power of our multi-agent paradigm compared with traditional multi-step optimization process **through additional experiments** in Appendix A.3.
>
> We sincerely appreciate your review and hope our responses, supplemented by additional experiments, address your concerns. Your insights are invaluable, and we would be grateful for further suggestions on how we might refine our work to warrant a higher evaluation.
>
> Looking forward to your feedback!
>
> Best regards,
>
> ICLR Author

---

> ### Author Response · Authors · 2024-12-03
>
> Dear Reviewer dSQq,
>
> Thank you for your valuable efforts in reviewing our work! We have thoroughly addressed all your concerns and questions, resolving the issues you raised regarding our paper. With the discussion period concluding in **9 hours**, we kindly request you to review our rebuttal at your earliest convenience.
>
> We look forward to your feedback!
>
> Best regards,
>
> ICLR Author

---

> ### Author Response · Authors · 2024-12-03
>
> Dear Reviewer dSQq,
>
> Thank you for your thoughtful and valuable feedback on our work! We have conducted additional experiments and made revisions to our paper to address the questions and concerns you raised. We hope these updates effectively address your points.
>
> If you have any further concerns, we will carefully address them in the final version of our paper.
>
> We kindly hope you take a moment to review our responses.
>
>
> Best regards,
>
> ICLR Authors

---

### Official Review · Reviewer_WfeZ · 2024-11-04

**Soundness:** 3
**Presentation:** 4
**Contribution:** 4
**Rating:** 8
**Confidence:** 4

**Summary:**

The paper introduces a novel multi-agent collaborative data selection mechanism aimed at enhancing data efficiency during the pretraining of large language models (LLMs). Recognizing that existing data selection methods often operate independently and may conflict with one another, the authors propose a framework where each data selection method functions as an independent agent. An agent console dynamically integrates the information from all agents throughout the training process. The agents adjust their weights based on reward signals derived from the model's performance on reference tasks. The framework is designed to flexibly and robustly combine various data selection strategies, such as data quality scoring, topic diversity, and domain information. Extensive experiments demonstrate that this multi-agent approach significantly accelerates convergence in LLM training and achieves an average performance gain of up to 10.5% across multiple benchmarks compared to state-of-the-art methods.

**Strengths:**

1. Introducing a multi-agent framework to collaboratively select pretraining data is a novel idea that addresses inherent conflicts among existing methods.
2. The empirical evaluation is extensive, comparing the proposed method against a wide range of baselines and demonstrating significant improvements.
3. The paper clearly articulates the motivation, methodology, and findings, making it accessible to readers.
4. Improving data efficiency in LLM pretraining is a critical challenge, and the proposed method offers a practical solution with demonstrable benefits.

**Weaknesses:**

1. While the experiments show promising results on models up to 1.3 billion parameters, it is unclear how the approach scales to larger models commonly used in practice.
2. The choice of agents (quality, domain, topic) seems somewhat ad-hoc. A discussion on how to generalize the selection of agents or include other data selection criteria would strengthen the paper.
3. While ablation studies are included, more detailed analysis on how each agent contributes to different types of tasks could provide deeper insights.

**Questions:**

1. How does the proposed framework perform when scaling up to larger models (e.g., 10B+ parameters) and datasets (e.g., trillions of tokens)?
2. Can you provide a more detailed analysis of the computational overhead introduced by the multi-agent system compared to baseline methods?
3. What guidelines can be provided for selecting or designing agents for other data selection criteria? Is the framework flexible enough to incorporate new agents easily? How sensitive is the method to the choice of number and types of agents?
4. How sensitive is the performance to the choice of reference tasks used for calculating rewards in the influence functions?
5. Can you elaborate on how the dynamic adjustment of agent weights impacts the learning process over time? Are there scenarios where this adjustment could lead to suboptimal data selection?

---

> ### Author Response · Authors · 2024-11-22
> **Authors Response 1/3**
>
> Thank you for the efforts in reviewing our paper!
>
> > W1: While the experiments show promising results on models up to 1.3 billion parameters, it is unclear how the approach scales to larger models commonly used in practice.
> >
>
> > Q1: How does the proposed framework perform when scaling up to larger models (e.g., 10B+ parameters) and datasets (e.g., trillions of tokens)?
> >
>
> To evaluate the scalability of our approach, we conducted additional experiments training a 3.6 billion parameter model from scratch based on the LLaMA 3.2 architecture, which further demonstrates the scalability of our method. *So far we have trained on 36 billion tokens and achieved strong performance, with plans to continue training with additional tokens according to scaling laws.* Note that when compared to random selection, our method shows consistent performance improvements across all downstream tasks, achieving a 13.7% increase in average accuracy—significantly higher than the 10.5% improvement observed with the 1.3B models.
>
> Due to computational limitations, we were unable to further scale to larger models (e.g., those with 10B+ parameters) within the tight time constraint. We want to gently point out that similar research about pretraining data selection is usually conducted on a similar or even smaller scale [1, 2, 3]; we plan to leave such large-scale experiments as important future work.
> On the other hand, based on trends across three different model sizes (373M, 1.3B, and 3.6B), our approach consistently outperforms random selection by over 10% on average. This consistent advantage makes us believe that it suggests our method has strong potential for training even larger models, including those with 10B+ parameters.
>
> |  | ARC-C | ARC-E | MathQA | MMLU | O.B.QA | SIQA | W.G. | C.S.QA | BoolQ | RACE | Average |
> | --- | --- | --- | --- | --- | --- | --- | --- | --- | --- | --- | --- |
> | 3.6B (Random) | 17.7 | 34.8 | 21.3 | 23.0 | 12.0 | 32.9 | 50.2 | 19.6 | 37.8 | 20.9 | 27.0 |
> | 3.6B (Ours) | **21.3** | **42.9** | **21.9** | **24.0** | **15.8** | **33.9** | **51.0** | **20.4** | **54.8** | **21.2** | **30.7** |
>
> **In our updated draft, we have added an analysis of the generalization of our methods in Appendix A.6.**
>
> [1] Engstrom L, Feldmann A, Madry A. Dsdm: Model-aware dataset selection with datamodels. ICML, 2024.
>
> [2] Wettig A, Gupta A, Malik S, et al. QuRating: Selecting High-Quality Data for Training Language Models. ICML, 2024.
>
> [3] Yu Z, Das S, Xiong C. MATES: Model-Aware Data Selection for Efficient Pretraining with Data Influence Models. NeurIPS, 2024.

---

> ### Author Response · Authors · 2024-11-22
> **Authors Response 2/3**
>
> > W2: The choice of agents (quality, domain, topic) seems somewhat ad-hoc. A discussion on how to generalize the selection of agents or include other data selection criteria would strengthen the paper.
> >
>
> > Q3: What guidelines can be provided for selecting or designing agents for other data selection criteria? Is the framework flexible enough to incorporate new agents easily? How sensitive is the method to the choice of number and types of agents?
> >
>
> Thank you for your valuable feedback.
>
> **Rationale behind current agent selection.** Our approach draws significant inspiration from the LLaMA 3.1 [1] Technical Report, which emphasizes data selection based on quality, domain, and topic. While the report does not provide a detailed framework, we utilized Fine-Web Edu and a topic classifier (trained on a large CC corpus, as shown in Figure 3) to define agents, forming the backbone of our data selection strategy.
>
> **Generalize with new criteria.** We appreciate the suggestion to explore how to generalize the agent selection methods or incorporate additional data selection criteria. In fact, our approach is designed to be flexible, allowing for the integration of new agents *as long as the new criteria can be divided into distinct subcategories and used to label the full dataset*. New rules can then be seamlessly incorporated into our agent collaboration framework. Specifically, adding a new agent to the framework involves the following steps:
>
> - Annotate a sampled dataset based on the new criterion and train a classifier for the criterion;
> - Define the new agent’s action space and memory;
> - Use the classifier to label the entire pretraining dataset;
> - Assign weights to the new agent and integrate it into the collaboration function;
> - Initialize the agent using regression strategies.
>
> By following these steps, new agents can be efficiently integrated into our framework. **We have revised our paper to provide an additional guidance in Appendix A.2.**
>
> **Sensitivity of agent number and types.** The collaborative mechanism dynamically adjusts the influence of each agent in the final decision-making process based on model preferences, ensuring that stronger agents have a greater impact while less significant agents are deprioritized. This design aims to optimize the role of each agent during collaboration. Consequently, in principle, the framework is not sensitive to the number or specific selection of agents. Our ablation studies further demonstrate that adding new agents generally enhances the performance of existing agents.
>
> [1] https://ai.meta.com/research/publications/the-llama-3-herd-of-models/
>
>
> > W3: While ablation studies are included, more detailed analysis on how each agent contributes to different types of tasks could provide deeper insights.
> >
>
> Based on our experiments, we provide a more detailed analysis of how the agent contributes to different types of tasks. Specifically, we summarize the following observation:
>
> 1. **Quality agent**: This agent primarily enhances performance in problem-solving tasks like ARC-E, ARC-C, and MathQA. These findings suggest that emphasizing quality (data with more educational knowledge) significantly benefits tasks that rely on the model’s inherent knowledge. However, its impact is less pronounced on tasks requiring domain-specific knowledge or contextual understanding, such as OpenBookQA, WineGrade, BoolQ, and RACE.
> 2. **Domain agent**: The domain agent contributes most to commonsense reasoning tasks, such as CommonsenseQA, and reading comprehension tasks, such as BoolQ. Since these tasks demand domain knowledge, incorporating a domain agent helps balance domain-specific information, thereby improving model performance on these tasks.
> 3. **Topic agent**: The topic agent is most effective in comprehensive tasks requiring knowledge across multiple topics, such as MMLU. Additionally, it provides significant contributions to commonsense reasoning tasks like SocialIQA and CommonsenseQA.
>
> **Following the suggestions from other reviewers, we scaled our model to a 1.3B parameter size, and the conclusions remain consistent. Additionally, we have revised our analysis for the ablation study in the paper and added a detailed analysis in Appendix A.5 to provide clearer and more detailed insights.**

---

> ### Author Response · Authors · 2024-11-22
> **Authors Response 3/3**
>
> > Q2: Can you provide a more detailed analysis of the computational overhead introduced by the multi-agent system compared to baseline methods?
> >
>
> Thank you for highlighting this critical point. We appreciate the chance to offer a more in-depth analysis and comparison. Below, we detail the computational overhead introduced by our methods in contrast to the two leading baselines, as shown in Table 1 of the paper. **We have updated our drafts with this detailed analysis in Appendix A.4.**
>
> **Offline labeling efficiency:** Our method involves a one-time dataset labeling process, requiring approximately $9.91 \times 10^{19}$ FLOPs using a 109M BERT-based model for inference. This is substantially more resource-efficient than Qu-Rating, which employs a 1.3B Sheared-LLaMA for inference and consumes $7.13 \times 10^{20}$ FLOPs.
>
> **Online update efficiency:** For adaptive online updates, both our approach and MATES compute influence scores with $1.19\times 10^{18}$ FLOPs. However, MATES involves labeling the entire dataset with a 109M BERT-based model in every round, amounting to $1.98 \times 10^{20}$ FLOPs across four data selection stages. In contrast, our method avoids re-labeling the entire dataset, significantly reducing the computational cost by focusing on labeling the large pretraining datasets only once.
>
> Overall, our approach cuts the computational cost in half compared to MATES and requires only about 1/7 of the computational resources used by Qu-Rating.
>
> | Seletion Method | Offline Computation Cost (FLOPs) | Online Computation Cost (FLOPs) | Overall Computation Cost (FLOPs) |
> | --- | --- | --- | --- |
> | QuRating[1] | $7.13 \times 10^{20}$ | N.A. | $7.13 \times 10^{20}$ |
> | MATES[2] | N.A. | $1.99 \times 10^{20}$ | $1.99 \times 10^{20}$ |
> | Multi-agent collaboration (ours) | $9.91 \times 10^{19}$ | $1.19\times 10^{18}$ | $1.00 \times 10^{20}$ |
>
> [1] Wettig A, Gupta A, Malik S, et al. QuRating: Selecting High-Quality Data for Training Language Models. In Forty-first International Conference on Machine Learning.
>
> [2] Yu Z, Das S, Xiong C. MATES: Model-Aware Data Selection for Efficient Pretraining with Data Influence Models. Advances in neural information processing systems, 2024.
>
> > Q4: How sensitive is the performance to the choice of reference tasks used for calculating rewards in the influence functions?
> >
>
> Thanks for pointing out this meaningful question. We performed an additional ablation study on the selection of reference tasks and present the results of a three-shot evaluation. **We have updated our drafts with this additional ablation study in Appendix A.5.2.**
>
> In our experiments, we observe that while the choice of reference tasks can influence performance, the impact on average performance is *relatively marginal* (within 0.5 points). Using different reference tasks consistently leads to a significant improvement in average performance compared to random data selection, demonstrating that our method is not sensitive to the choice of reference tasks.
>
> |  | ARC-C | ARC-E | MathQA  | MMLU | O.B.QA | SIQA | W.G.  | C.S.QA | BoolQ | RACE | Average |
> | --- | --- | --- | --- | --- | --- | --- | --- | --- | --- | --- | --- |
> | LAMBADA&SQuAD&Jeopardy | **31.5** | **65.8** | 23 | 26.6 | 24.6 | 39.9 | 54.1 | 20.1 | **60.4** | **30.5** | **37.7** |
> | LAMBADA | 31.2 | 64.3 | 22.3 | **26.8** | 23.5 | 39.6 | **54.6** | 20.4 | 59.6 | 30.1 | 37.2 |
> | SQuAD | 30.9 | 65.1 | 23.4 | 25.9 | **24.9** | 40.1 | 53.8 | 21.2 | 59.1 | 29.3 | 37.4 |
> | Jeopardy | 30.3 | 63.9 | **23.6** | 26.3 | 24.1 | **40.7** | 54.5 | **21.8** | 59.1 | 30.2 | 37.5 |
> | Random selection | 23 | 54.6 | 22.1 | 24.9 | 18.8 | 40.3 | 52.9 | 21.5 | 53 | 29.8 | 34.1 |
>
> > Q5: Can you elaborate on how the dynamic adjustment of agent weights impacts the learning process over time? Are there scenarios where this adjustment could lead to suboptimal data selection?
> >
>
> Our adaptive adjustment of agent collaboration weights significantly enhances the learning process over time. As shown in the Ablation Study in Table 2, we compared our approach (utilizing dynamic agent collaboration weights), to a truncated version employing fixed collaboration weights. Our dynamic adjustment method consistently outperforms the fixed-weight approach, delivering an overall performance improvement of 7.0%. The detailed results for each training stage are presented in the following table.
>
> By applying the dynamic agent collaboration score adjustment outlined in Equation 9, we adjust the impact of each agent based on the model's preferences, optimizing the collaborative strategy for individual agents. This approach in principle prevents suboptimal data selection, and our experiments revealed no instances of suboptimal outcomes.
>
> | Training steps | 1500 | 3000 | 4500 | 6000 | 7500 |
> | --- | --- | --- | --- | --- | --- |
> | with dynamic adjustment | 31.1 | **33.3** | **35.9** | **36.7** | **37.7** |
> | without dynamic adjustment | 31.1 | 32.9 | 33.6 | 34.4 | 35.1 |

---

> ### Author Response · Authors · 2024-11-25
>
> Dear Reviewer WfeZ,
>
> We sincerely appreciate the time and effort you have dedicated to reviewing our work!
>
> To address your feedback on generalization and reference task selection, we’ve added new experiments, detailed in Appendices A.5 and A.6. We’ve also clarified points on agent selection, agent contributions, guidelines for adding new agents, and analysis on efficiency in the revised paper, with further details in Appendix A.4. We hope that our response, along with the additional experiments and the revised draft, has effectively addressed your questions and concerns.
>
> As the discussion period nears its conclusion, we would greatly appreciate it if you could review our reply at your convenience. If there are any unresolved concerns, we are fully committed to addressing them promptly and to the best of our ability.
>
> We are deeply grateful for your patience and insightful contributions, and we sincerely look forward to further discussion!

---

> ### Author Response · Authors · 2024-12-02
>
> Dear Reviewer WfeZ,
>
> Thank you for your valuable efforts in reviewing our work! As the discussion period approaches its conclusion, we kindly hope you can spare some time to review our rebuttal.
>
> To address the concerns and questions you raised:
>
> + *Scalability of our methods*: We conducted **additional experiments on 3.6B and 8B models**, expanding our analysis across four model sizes (373M, 1.3B, 3.6B, and 8B parameters). Our findings consistently show over a 10% average performance improvement compared to random selection across a range of widely-used benchmarks, demonstrating the scalability, generalizability, and potential impact of our methods.
>
> + *Choice of agent and agent ability*: We **revised our analysis** on agent selection experiments and **added further details** in Appendix A.5.
>
> + *Computational overhead*: A **comparison of computational overhead with other SOTA methods** is now included in Appendix A.4.
>
> + *Guidelines for adding new agents*: We provided **step-by-step instructions** for adding new agents in Appendix A.2.
>
> + *Sensitivity to reference tasks*: **Additional experiments** exploring the sensitivity of reference tasks have been added in Appendix A.5.2.
>
> + *Dynamic adjustments*: We included **detailed experimental results** and plan to provide additional analysis in the final paper.
>
> We sincerely thank you again for reviewing our work and hope our responses, along with the supplementary experiments, adequately address your concerns. We welcome any further suggestions for improvement and look forward to your feedback!
>
> Best regards,
>
> ICLR Author

---

> > ### Comment · Reviewer_WfeZ · 2024-12-02
> >
> > Thank you for your detailed and thoughtful responses to my comments. Your clarifications have addressed all of my concerns, and I now have a much clearer understanding of the points in question. The revisions you made significantly improve the clarity and rigor of the paper, and I truly appreciate the effort you've put into this.

---

> ### Author Response · Authors · 2024-12-02
>
> Dear Reviewer WfeZ,
>
> Thank you for your kind and encouraging response! We sincerely appreciate the thoughtful concerns and questions you raised, which have been invaluable in helping us refine and enhance our work! We are committed to continuously improving it for the final version.
>
> Best regards,
>
> ICLR Authors

---

### Official Review · Reviewer_wvnx · 2024-11-10

**Soundness:** 2
**Presentation:** 2
**Contribution:** 2
**Rating:** 5
**Confidence:** 3

**Summary:**

This paper proposes to a mechanism to select data into the training process based on 3 main measure (quality, domain and topic) of the data. The 3 measures are adjusted dynamically and aggregating together to determine whether data point can be selected during the training process. A RL paradigm is employed to realize the proposal. Good performance is demonstrated in the provided experiments.

**Strengths:**

• A good analysis of how different aspects influence the performance of LLM training.
• The experiment seems to demonstrate this kind of scoring of data points can help to improve the data deficiency and performance.

**Weaknesses:**

• The connection to agent or multi-agent paradigm seems weak to me. It might not be necessary to formulate the problem and the solution via "agent" concept. A direct stochastic optimization formulation might provide  more direct description and help audience better mastering what is the proposal.

**Questions:**

1. Is it possible to not employ the agent or multi-agent metaphor to formulate the proposal? What is the truth power of the proposal? Is it a compossible and multi-step paradigm of stochastic optimization  based on 3 hand-crafted dimensions?
2. Ine line 161, what is the loss function l? Also please explain more regarding the definition of the reward function. How is it different o related to the loss function of LLM auto-regression loss or other kinds of losses (if any other).
3. In Algorithm 1, please clarify whether the sampling distribution of the data are different from iteration to iteration (line 3).
4. In line 275, 373M model is used in ablation study. It is pretty small a model. The conclusions drawn from it might not be transferrable to LLMs of billions of parameters. Please justify the study.

---

> ### Author Response · Authors · 2024-11-22
> **Authors Response 1/3**
>
> Thank you for the efforts in reviewing our paper!
>
> > W1: The connection to agent or multi-agent paradigm seems weak to me. It might not be necessary to formulate the problem and the solution via "agent" concept. A direct stochastic optimization formulation might provide more direct description and help audience better mastering what is the proposal.
> >
>
> > Q1: Is it possible to not employ the agent or multi-agent metaphor to formulate the proposal? What is the truth power of the proposal? Is it a compossible and multi-step paradigm of stochastic optimization based on 3 hand-crafted dimensions?
> >
>
> Thank you for your insightful comments! To clarify the problem's formulation, we have updated and refined its description in our paper (highlighted in blue) and provided detailed mathematical formulations in Appendix A.3. Below, we outline the distinctions between our approach and stochastic optimization methods:
>
> We want to emphasize that while our method is based on the optimization problem, the core strength of our approach lies in the agent design. In fact, **the "truth power" of the agent arises from the dynamic adaptation of both the individual agent's weights and the collaborative weights shared among multiple agents, which cannot be directly showcased through a straightforward optimization paradigm.** Our experiments demonstrate that this dynamic weight adjustment throughout the training process leads to superior results, especially when compared to approaches using fixed weights, no collaboration, competitive methods, or single-agent strategies.
>
> We further compare our multi-agent paradigm with the traditional optimization paradigm to highlight the differences between the two approaches.
> In the multi-agent collaboration framework we propose, the final decision-making is based on Equation 7, $S(x_i) = \sum_{\mathcal{A}\in set(\mathcal{A})}\theta\_{\mathcal{A}}\cdot S\_{\mathcal{A}}(x\_i)$, to select the top-k scored data, where each agent provides a score $S\_{\mathcal{A}}(x_i)$, and the agent console records $\theta_{\mathcal{A}}$, which represents the contribution of each agent in the collaboration. We consider three possible cases for our framework, comparing its relationship with traditional optimization problem.
>
> 1. *Single-agent case:* If only one agent is involved, $\theta$ becomes irrelevant, reducing the problem to a classical optimization scenario where the agent greedily selects the optimal data based on one criteria.
> 2. *Multi-agent competitive mechanism:* When multiple agents are present, $\theta$ reflects each agent’s capability. Selecting the best-performing agent for decision-making introduces a heuristic competitive mechanism, building upon the classical optimization framework.
> 3. *Multi-agent collaborative mechanism:* Alternatively, when multiple agents are involved, $\theta$ can be used to weigh each agent's contributions for decision-making. This introduces a smoother heuristic cooperative mechanism, **extending the classical optimization framework by leveraging weighted collaboration.** This heuristic cooperative mechanism dynamically adjusts the influence of each agent based on the model's current preferences, enabling more effective data filtering decisions.
>
> In practice, we choose to use the multi-agent collaborative mechanism for data selection. We have added comparisons with single-agent and competitive mechanisms in **Appendix A.3** to further elaborate the effectiveness of collaboration.
>
> **Since the traditional multi-step optimization paradigm cannot fully explain why our method truly works, we chose not to use it to demonstrate our framework. Instead, we adopted the multi-agent paradigm to highlight the dynamic adjustment of agent weights and the collaborative weight adjustments among different agents, showcasing the true strengths of our framework.**

---

> ### Author Response · Authors · 2024-11-22
> **Authors Response 2/3**
>
> > Q2: In line 161, what is the loss function l? Also please explain more regarding the definition of the reward function. How is it different o related to the loss function of LLM auto-regression loss or other kinds of losses (if any other).
> >
>
> To address your concerns, **we have clarified these definitions and expanded on the explanation in Section 3.1, with the updates highlighted in blue for better visibility.** We enumerate the updates here:
>
> **Definitions of loss function and reward function:** We follow the definition of this loss function as stated in Dsdm [1] to define the loss function.
>
> - The loss function in Equation 1 is defined as the trained model population loss as $\mathcal{L}(\mathcal{D}_k\mid \mathcal{M},\mathcal{T}\_{\text{eval}}) \coloneqq \mathbb{E}\_{x \sim \mathcal{T}\_{\text{eval}}} \left[\ell(x; \mathcal{O}(\mathcal{M},\mathcal{D}\_k))\right]$, where $\ell(x; \mathcal{O}(\mathcal{M},\mathcal{D}_k))$ denotes the cross-entropy loss for model $\mathcal{M}$ on example $x$. The expectation in the population loss is over downstream tasks. While the downstream tasks are unknown during training process, the loss function can not be optimized directly. Therefore, reference tasks are introduced to estimate the trained model population loss.
> - The reward function is defined as an extension of the loss function $R(\mathcal{D}\_k \mid \mathcal{M}, \mathcal{T}\_{\text{ref}}) \coloneqq \mathbb{E}\_{x \sim \mathcal{T}\_{\text{ref}}} \left[-\ell(x; \mathcal{O}(\mathcal{M},\mathcal{D}\_k))\right]$, where it is the expection of negative population loss estimated over reference tasks. Intuitively, a higher reward indicates better model performance.
>
> **Comparison to LLM auto-regression loss:** The primary distinction between the loss in LLM auto-regression and the loss defined in our work lies in their objectives. While the LLM auto-regression loss is designed to optimize the language model parameters during training on the training set, our defined loss function targets to optimize the model's performance for downstream tasks throughout the training process. Since downstream tasks are unknown during the training phase, reference tasks are used to estimate this loss.
>
> [1] Logan Engstrom, Axel Feldmann, and Aleksander Madry. Dsdm: Model-aware dataset selection with datamodels. In Forty-first International Conference on Machine Learning.
>
> > Q3: In Algorithm 1, please clarify whether the sampling distribution of the data are different from iteration to iteration (line 3).
> >
>
> We want to clarify that *our sampling distribution remains consistent across iterations***.** Concretely, at each iteration of data selection, our goal is to sample a subset of data that can represents the entire data distribution. As the overall data distribution remains fixed, the sampled dataset can be considered a fixed distribution. **We have revised the paper in blue color to explicitly clarify these details in Algorithm 1 and emphasize the fixed nature of the sampling distribution across iterations.**

---

> ### Author Response · Authors · 2024-11-22
> **Authors Response 3/3**
>
> > Q4: In line 275, 373M model is used in ablation study. It is pretty small a model. The conclusions drawn from it might not be transferrable to LLMs of billions of parameters. Please justify the study.
> >
>
> We appreciate the suggestion and further conduct all the ablation study on 1.3B models, and here are the results. Compared to the ablation study conducted on the 373M model, **the overall agent performance remains consistent**. While the performance ranking of individual agents shifts slightly, the contribution of each agent to different types of tasks remains steady. The following table illustrate the results over the 1.3B models:
>
> |  | ARC-C | ARC-E | MathQA | MMLU | O.B.QA | SIQA | W.G. | C.S.QA | BoolQ | RACE | Average |
> | --- | --- | --- | --- | --- | --- | --- | --- | --- | --- | --- | --- |
> | Quality&Domain&Topic Agent | **31.5** | **65.8** | **23** | 26.6 | **24.6** | 39.9 | **54.1** | 20.1 | **60.4** | 30.5 | **37.7** |
> | without collaboration update | 26.3 | 59.4 | 21.3 | 25.1 | 20.5 | 38.9 | 52.9 | 19.8 | 58.1 | 28.3 | 35.1 |
> | Domain&Quality Agent | 29.7 | 63.3 | 22.6 | 25.1 | 21.8 | **40.5** | 53.1 | 20.3 | 59.5 | 28.8 | 36.5 |
> | Topic&Quality Agent | 28.1 | 62.9 | 22.3 | 26.5 | 22.6 | 39.6 | 51.8 | **21.7** | 56.7 | **30.7** | 36.3 |
> | Domain&Topic Agent | 25.2 | 55.6 | 21.8 | 26.5 | 23.1 | 39.1 | 53.7 | 20.9 | 57.5 | 29 | 35.2 |
> | Quality Agent | 29.7 | 59.1 | 22.4 | 25.3 | 21.1 | 38.5 | 51.2 | 19.1 | 57.2 | 28.3 | 35.2 |
> | Domain Agent | 25.6 | 54.1 | 21.4 | 25.9 | 22.3 | 38.1 | 53.6 | 20 | 58.1 | 27.9 | 34.7 |
> | Topic Agent | 25.3 | 55.3 | 21.9 | **27.1** | 22.1 | 39.4 | 51.5 | 19.8 | 56.3 | 28.9 | 34.8 |
> | No Agent | 23 | 54.6 | 22.1 | 24.9 | 18.8 | 40.3 | 52.9 | 21.5 | 53 | 29.8 | 34.1 |
>
> **This result has been incorporated into the revised paper with a detailed analysis. Additionally, we have included Appendix A.5, which presents the ablation study on both the 373M and 1.3B models**, providing further discussion on the results and the contributions of each agent to various task types.

---

> ### Author Response · Authors · 2024-11-25
>
> Dear Reviewer wvnx,
>
> We sincerely appreciate the time and effort you have dedicated to reviewing our work!
>
> To address your concerns regarding the agent formulation, the definition of the loss and reward function, and the algorithm, we have revised our draft for clarity and included additional explanations in Appendix A.3. Additionally, in response to your feedback about the ablation study, we have expanded it to include 1.3B models. We hope these updates, along with the revised draft and new experiments, effectively address your questions and concerns.
>
> As the discussion period draws to a close, we would be truly grateful if you could take a moment to review our response at your convenience. If there are any outstanding concerns that we have not yet addressed to improve our score, please let us know, and we will make every effort to resolve them promptly.
>
> We are deeply grateful for your patience and insightful contributions, and we sincerely look forward to further discussion!

---

> ### Author Response · Authors · 2024-12-02
>
> Dear Reviewer wvnx,
>
> We deeply appreciate your invaluable efforts in evaluating our work! As the discussion period is nearing its end, we kindly hope you can find time to review our rebuttal.
>
> Regarding the concerns and questions you raised:
>
> + *Multi-agent paradigm*:
> We have carefully revised our draft to clarify the concept and definition. Additionally, to address your concerns regarding the validity of the proposal and the distinction between our multi-agent paradigm and an optimization paradigm, we included **Appendix A.3, which provides a mathematical comparison**. Furthermore, we have conducted **additional experiments to demonstrate the advantages** of multi-agent collaboration compared to competition and collaboration without dynamic adjustments in Appendix A.3.
>
> + *Loss functions*:
> The formulation of the loss functions has been **clarified in the revised draft**, along with an expanded explanation to ensure greater comprehensibility.
>
> + *Algorithm 1*:
> We have **refined the draft** to make the sampling methods in Algorithm 1 more transparent and accessible.
>
> + *Ablation study*:
> The model size in all ablation studies has been extended to **match the 1.3B parameter models** used in the main experiments, which are four times larger than the original 373M models. The results consistently support the main conclusions drawn from smaller models. We have also updated the experimental sections and **Appendix A.5** to include further discussions.
>
> We sincerely thank you again for reviewing our work and hope that our responses, along with the additional experiments, address your concerns. We would greatly value any further suggestions for improving our work. We would greatly appreciate your insights on how we might **further refine our work to merit a higher evaluation from you**.
>
> Looking forward to your feedback!
>
> Best regards,
>
> ICLR Author

---

> ### Author Response · Authors · 2024-12-03
>
> Dear Reviewer wvnx,
>
> Thank you for your valuable efforts in reviewing our work! We have thoroughly addressed all your concerns and questions, resolving the issues you raised regarding our paper. With the discussion period concluding in **9 hours**, we kindly request you to review our rebuttal at your earliest convenience.
>
> We look forward to your feedback!
>
> Best regards,
>
> ICLR Author

---

> ### Author Response · Authors · 2024-12-03
>
> Dear Reviewer wvnx,
>
> Thank you for your thoughtful and valuable feedback on our work! We have conducted additional experiments and made revisions to our paper to address the questions and concerns you raised. We hope these updates effectively address your points.
>
> If you have any further concerns, we will carefully address them in the final version of our paper.
>
> We kindly hope you take a moment to review our responses.
>
>
> Best regards,
>
> ICLR Authors

---

### Author Response · Authors · 2024-11-22
**Global Response**

### **Overall merits:**

We thank all reviewers for their valuable comments! The reviewers acknowledge our work as a novel and practical solution to improving data efficiency in LLM pretraining using a multi-agent framework for collaborative data selection. They commend the innovation, extensive experiments that shows significant improvements, and clear presentation of motivation, methodology, and findings.

### **Current concerns:**

The concerns about the current draft mainly lie in three aspects:

- The definition and formulation of our multi-agent framework in the context of data selection (raised by *Reviewer wvnx, dSQq*, and primarily by *Reviewer 6r7V*);
- The computational costs and latency of our methods compared to baseline approaches (raised by *Reviewer WfeZ and dSQq*);
- The need for additional experiments to demonstrate the generalization and robustness of our methods in the main experiments and ablation studies (raised by *Reviewer wvnx, WfeZ and dSQq*).

In order to resolve these three issues, we have made the following efforts:

- **Enhanced framework explanation**: We have updated the technical discussion in the revised draft to clarify the origin and definition of our agent. Additionally, we have included supplementary mathematical formulations in Appendix A.3 to provide a clearer comparison of our multi-agent framework with other multi-agent formulations (such as multi-agent reinforcement learning) and traditional optimization problems.
- **Computational efficiency analysis**: We have calculated the overall computational Flops for both our methods and baseline approaches to highlight the efficiency of our methods.
- **Extended experiments**: We performed all feasible experiments suggested by the reviewers within our current computational constraints. Specifically, we: (1) expand all ablation studies from 373M to 1.3B models to address *Reviewer wvnx*'s concerns; (2) add experiments with 3.6B and 8B models to demonstrate the generalizability of our methods, addressing feedback from *Reviewer WfeZ and dSQq*; (3) conduct an ablation study on the selection of reference tasks to highlight the robustness of our methods, as requested by *Reviewer WfeZ*; and (4) included comparisons with open-source models around 1B parameters to further emphasize the potential of our approach, addressing *Reviewer dSQq*'s concerns.

We have updated the draft and appreciate it if reviewers would gently check the updated version of our paper.

---

> ### Author Response · Authors · 2024-11-26
> **Updating results from pretraining on 3.6B models**
>
> We would like to thank all the reviewers once again for their dedicated work in reviewing our work! According to the suggestion from reviewers, we have extended our work training on **3.6B LLaMA3.2** architecture models for more steps, and here are the results for training models from scratch using **60B tokens, taking approximatley 3.5 days training on 32 A100 GPUs for one model.** Our methods achieve **13.1% performance gains** in average.
>
> |  | ARC-C | ARC-E | MathQA | MMLU | O.B.QA | SIQA | W.G. | C.S.QA | BoolQ | RACE | Average |
> | --- | --- | --- | --- | --- | --- | --- | --- | --- | --- | --- | --- |
> | 3.6B (Random) | 24.2 | 53.7 | 22.1 | 23.4 | 18.6 | 36.7 | 53.0 | 19.6 | 41.0 | 21.4 | 31.4 |
> | 3.6B (Ours) | **29.4** | **64.0** | **24.0** | **24.8** | **22.6** | **37.5** | **54.3** | **20.7** | **54.1** | **23.8** | **35.5** |
>
> **As the discussion period has been extended, currently we are training 8B LLaMA architecture models from scratch** to further explore the scalability of our methods. We will continually update our latest results in the response windows.
>
> Thank you once again for your patience and insightful feedback! We are looking forward to further discussion!

---

> ### Author Response · Authors · 2024-11-30
> **Updating results from pretraining on 8B models**
>
> We sincerely thank all the reviewers for their invaluable efforts in evaluating our work! In our current setup, **training an 8B model from scratch on 7B tokens requires approximately one day using 32 A100 GPUs**. Due to computational limitations, we trained the 8B models from scratch for 6000 steps, corresponding to approximately 25.1B tokens. **Our results show that our methods outperform random selection approaches by 10.2%**, underscoring the scalability and effectiveness of our approach.
>
> As the discussion period concludes in three days, **we kindly request you to review our rebuttal and share any further concerns or suggestions that could help elevate the score.** Your time and feedback are greatly appreciated!
>
> |  | ARC-C | ARC-E | MathQA | MMLU | O.B.QA | SIQA | W.G. | C.S.QA | BoolQ | RACE | Average |
> | --- | --- | --- | --- | --- | --- | --- | --- | --- | --- | --- | --- |
> | 8B (Random) | 22.2 | 53.3 | 21.5 | 23.3 | 19.4 | 36.1 | 51.0 | 18.2 | 48.0 | 21.3 | 31.4 |
> | 8B (Ours) | **25.5** | **58.0** | **23.2** | **24.1** | **21.6** | **38.8** | **53.1** | **20.8** | **57.8** | **22.9** | **34.6** |

---

> ### Author Response · Authors · 2024-12-01
> **Addressing the most controversial points of our work**
>
> We sincerely thank all the reviewers for their invaluable efforts in evaluating our work. We also extend our deepest gratitude to the Area Chairs, Senior Area Chairs, and Program Chairs for facilitating the discussion period and ensuring a fair and thorough evaluation process.
>
> As the discussion period is set to conclude in two days and we have not yet received feedback from most of the reviewers, we would like to further address two most controversial points of our work raised during the reviews:
>
> + *Scalability of our methods (raised by Reviewer WfeZ and dSQq):*
> To address concerns about scalability, we have conducted additional experiments using 3.6B and 8B parameter LLaMA architecture models, complementing our original evaluations. This expands our analysis **across four different model sizes (373M, 1.3B, 3.6B, and 8B parameters)** and includes comparisons across various versions of LLaMA architectures (LLaMA2 and LLaMA3.2). Our results demonstrate that **our methods consistently achieve more than a 10% average performance improvement** compared to random selection across a range of widely-used benchmarks. We believe these findings provide strong evidence of the scalability, generalizability, and potential impact of our methods.
>
> + *Use of multi-agent to illustrate our method (raised by Reviewer wvnx, dSQq, and primarily by Reviewer 6r7V)*:
> Reviewer 6r7V appears to focus largely on the conceptual framework of our paper, specifically questioning the use of "agent" in describing our method. **The term "agent" in our work is derived from the classic definition in Professor Stuart J. Russell and Doctor Peter Norvig's highly regarded book Artificial Intelligence: A Modern Approach [1]**, where an intelligent agent is defined to "implement a function that maps percept sequences to actions."
> For clarity, we have designed our data selection agent based on this definition, providing detailed explanations throughout Section 3 and accompanying figures. Furthermore, we include a mathematical comparison with "multi-agent RL" in Appendix A.3 and carefully revise our draft to address conceptual concerns. We want to highlight that **our "agent" adheres to established concept from classical AI literature, avoiding the introduction of unconventional terminology**, and we ensure that the concept is used appropriately and is clearly defined. While we understand the potential for varied interpretations of "agent" across domains, we also want to emphasize that **the core methodologies and experimental results—the primary contributions of our work—are unaffected by the terminology used.** As emphasized by the reviewers, our work thoroughly examines the inherent conflicts present in existing methods (as noted by Reviewer wvnx, WfeZ, dSQq, and 6r7V), and our approach introduces a novel perspective and an innovative solution, utilizing multiple well-defined agents to tackle the critical challenge of data-efficient training for large language models (highlighted by Reviewer WfeZ and dSQq). We welcome any further discussion to refine and clarify our terminology.
>
> Reviewer 6r7V's emphasis on conceptual issues highlights their keen interest in the conceptual framework of our work. We appreciate this perspective and are committed to addressing any concrete concerns to enhance the clarity and accessibility of our paper.
>
> Once again, we thank all reviewers for their thoughtful evaluations of our work! We sincerely hope you find time to review our rebuttal and share any additional concrete concerns or suggestions that could help **strengthen our work and improve our score**.
>
> [1] Russell S J, Norvig P. Artificial intelligence: a modern approach[M]. Pearson, 2016.

---

> ### Author Response · Authors · 2024-12-02
> **Looking forward to your feedback on our rebuttal**
>
> Dear Reviewers,
>
> Thank you for your valuable time and thoughtful reviews of our work!
>
> As today marks the **final day** for author-reviewer discussions, we kindly request you to review our rebuttal at your earliest convenience.
>
> We have provided a **comprehensive summary** of the additional experiments and revisions made to address each of your concerns in our latest response for all the reviewers. We would greatly appreciate any further feedback or suggestions on **how we might improve our work to merit a higher evaluation.**
>
> We look forward to hearing your insights!
>
> Best regards,
>
> ICLR Authors

---

> ### Author Response · Authors · 2024-12-03
> **Looking forward to your feedback on our rebuttal**
>
> Dear Reviewers,
>
> We sincerely appreciate your time and the thoughtful feedback you have provided on our work.
>
> With only **6 hours remaining** in the author-reviewer discussion period, we kindly ask you to review our rebuttal at your earliest convenience. We would be grateful for any additional feedback or suggestions to further enhance the quality of our work.
>
> We look forward to hearing your insights!
>
> Best regards,
>
> ICLR Authors

---

> ### Author Response · Authors · 2024-12-03
> **Explain once again the agent design for Reviewer 6r7Vr due to the quiet revision of review after author-reviewer discussion period**
>
> Although *Reviewer 6r7V* acknowledged our familiarity with the book that provides a comprehensive introduction to intelligent agents [1], we found that **Reviewer 6r7Vr quietly revised the original review after the author-reviewer discussion period**, continuing to *repeat the original questions and weaknesses* that our agents are not true agents **without offering concrete justification**. We believe it is important to **concretely** explain the foundation and rationale behind our agent design once again for this reviewer.
>
> We acknowledge that our initial submission lacked sufficient detail about the basis and rationale for our agent design, and how it differs from agents in "multi-agent RL" (mentioned by *Reviewer 6r7V*), traditional optimization problem (mentioned by *Reviewer wvnx*) and multi-step process (mentioned by *Reviewer dSQq*). We extend our apologies to all the reviewers for **this writing problem.** To address this, **we have carefully revised our paper, incorporating precise mathematical definitions of the agent's design, functionality, and its distinctions from agents in "multi-agent RL"(Appendix A.3.1), multi-step optimization problem (Appendix A.3.3) in Appendix A.3. We hope that the current version presents a clear and comprehensive explanation of our agents' design and operation.**
>
> Specifically, as stated in Chapter 2 of the book [1]:
>
> *"An agent is anything that can be viewed as perceiving its environment through sensors and acting upon that environment through actuators."*
>
> Our agent was designed based on this definition and the accompanying explanations in the text. It closely resembles the model-based reflex agent. We provide a concrete explanation of our agent using the pseudocode for the **model-based reflex agent from Section 2.4 (Page 53 of [1]):**
>
>
> ```jsx
> function MODEL-BASED-REFLEX-AGENT(percept) returns an action
> persistent:
> STATE, the agent’s current conception of the world state // The internal state is the current memory and weight of each agent.
> TRANSITION MODEL, a description of how the next state depends on the current state and action // This is how the agent updates its memory and weight, which is defined by Equation 5 in our work.
> SENSOR MODEL, a description of how the current world state is reflected in the agent’s percepts // In our work, this perception process is structured into several steps: (1) sampling data, (2) interacting with the current model to score the sampled data, and (3) receiving feedback from the environment. These steps define how the agent perceives the current state of the world. After perceiving the current state, the data selection agent updates its internal memory and weights accordingly.
> RULES, a set of condition–action rules // Which is the predefined data selection criterion for each agents.
> ACTION, the most recent action, initially none  // The data selection agent prioritize the good data according to its rules and states.
> state ← UPDATE-STATE(state, action, percept, transition model, sensor model)
> rule ← RULE-MATCH(state, rules)
> action ← rule.ACTION
> return action
> ```
>
> At each data selection stage, our data selection agent adheres to the function outlined above to take action. It observes the global state (the current model state) and updates its internal states accordingly, enabling it to prioritize high-quality data based on its rules and internal state. **The workflow of our designed data selection agent closely mirrors the workflow of the model-based reflex agent described in this book.**
>
> The algorithm above clearly demonstrates how our agent adheres to the agent definition provided in Chapter 2 of [1]. *We have included a similar explanation in the revised version of Appendix A.3 of our paper.*
>
> **We believe this additional explanation as well as our revised draft could show that our "agent" adheres to established concept from classical AI literature, avoiding the introduction of unconventional terminology, and we ensure that the concept is used appropriately and is clearly defined.**
>
> [1] Russell S J, Norvig P. Artificial intelligence: a modern approach[M]. Pearson, 2016.

---

### Author Response · Authors · 2024-11-30

Dear ICLR 2025 Area Chairs, Senior Area Chairs, and Program Chairs,

Wishing you a joyful and blessed Thanksgiving!

We deeply appreciate the time and effort you invest in overseeing the review process of our paper. Your dedication and critical role in ensuring a fair and thorough evaluation mean a great deal to us.

As the discussion period approaches its conclusion in three days, we have received only one response from a reviewer to our rebuttal. While we appreciate their input, **the feedback primarily addresses writing issues without raising substantive concerns and seems to overlook the contributions of our methodological designs and comprehensive experiments. We believe that more detailed and constructive feedback would greatly enrich the discussion.**

We kindly request your assistance in reaching out to the reviewer and ensuring an impartial and balanced assessment of our work.

Thank you once again for your support.

Best regards,
ICLR Authors

---

### Comment · Senior_Area_Chairs · 2024-11-30
**Your response to authors' rebuttal needed ASAP**

Dear Reviewers,

*Would you please respond to the authors' rebuttal ASAP?* We are drawing close to the end of the author-reviewer discussion.

*Reviewer 6r7V*: Would you please provide some justifications to the authors why "their answers do not change my original evaluation"?

Many thanks for your reviewing effort!

Your SAC

---

### Meta-Review · Area_Chair_2EiV · 2024-12-22

**Metareview:**

This paper studies the data selection problem for pretraining large language models (LLMs). A data selection rule based on the weighted score of multiple criteria has been proposed, and the algorithm selects the data based on the weighted scores. The reviewers' comments are mixed. Most of the reviewers are concerned about the writing and the language of this paper as the word "Multi-Agent" often refers to the literature in "Multi-Agent Reinforcement Learning", but in this paper, there seem no dynamics involved, so may not necessarily need to define the RL problem. Another issue is on the scale of the simulation, the current simulations are on relatively small models, and extending to 8B LLMs seems a new norm. Therefore, I would suggest authors revise the paper based on the above comments and resubmit it in the next conference.

**Additional Comments On Reviewer Discussion:**

Most of the reviewers participated in the discussion except one reviewer wvnx, but her comments on the small scale of the simulation are critical. This paper is an empirical paper on LLM, so the evaluation of a 1.3B LLM does not meet the expectation.

---

### Decision · Program_Chairs · 2025-01-22

Reject